# Effective light cone and digital quantum simulation of interacting bosons

Tomotaka Kuwahara [1,2,3] ✉, Tan Van Vu [1] & Keiji Saito[4]

The speed limit of information propagation is one of the most fundamental features in non-equilibrium physics. The region of information propagation by finite-time dynamics is approximately restricted inside the effective light cone that is formulated by the Lieb-Robinson bound. To date, extensive studies have been conducted to identify the shape of effective light cones in most experimentally relevant many-body systems. However, the Lieb-Robinson bound in the interacting boson systems, one of the most ubiquitous quantum systems in nature, has remained a critical open problem for a long time. This study reveals a tight effective light cone to limit the information propagation in interacting bosons, where the shape of the effective light cone depends on the spatial dimension. To achieve it, we prove that the speed for bosons to clump together is finite, which in turn leads to the error guarantee of the boson number truncation at each site. Furthermore, we applied the method to provide a provably efficient algorithm for simulating the interacting boson systems. The results of this study settle the notoriously challenging problem and provide the foundation for elucidating the complexity of many-body boson systems.

Causality is a fundamental principle in physics and imposes the strict prohibition of information propagation outside light cones. The non-relativistic analog of causality was established by Lieb and Robinson[1], who proved the existence of the effective light cone. The amount of information outside the light cone decays exponentially with the distance. The recent experimental developments have allowed one to directly observe such effective light cones in various experimental setups[2–4]. The Lieb-Robinson bound provides a fundamental and universal speed limit (that is, the Lieb-Robinson velocity) for non-equilibrium structures in real-time evolutions. Furthermore, the Lieb-Robinson bound also offers critical insights into the steady states and spectral properties of the systems using the Fourier transformation. In the past decades, the Lieb-Robinson bound has found diverse applications in interdisciplinary fields, such as the area law of entanglement[5,6], quasi-adiabatic continuation[7], fluctuation theorem for pure quantum states[8], clustering theorems for correlation functions[9–11], tensor-network based classical simulation of many-body

systems[12,13], optimal circuit complexity of quantum dynamics[14], sample complexity of quantum Hamiltonian learning[15], and quantum information scrambling[16]. Owing to these crucial applications, the Lieb-Robinson bound has become a central topic in the field of quantum many-body physics.

Lieb and Robinson argued that the speed of information propagation is finitely bounded; that is, the effective light cone is linear with time. One might have a naive expectation that this is true in generic quantum many-body systems. However, to justify such an intuition, we must assume the following conditions: (a) the interactions are short-range, and (b) the strength of interactions is finitely bound. Understanding the breakdown of the two above-mentioned conditions is inevitable for comprehensively describing the information propagation in all experimentally relevant quantum many-body systems. The breakdown of condition (a) should be easy to imagine. Under long-range interactions, the information propagates immediately to an arbitrarily distant point, causing one to intuitively assume that the

[1]Analytical quantum complexity RIKEN Hakubi Research Team, RIKEN Center for Quantum Computing (RQC), Wako, Saitama 351-0198, Japan. [2]RIKEN Cluster for Pioneering Research (CPR), Wako, Saitama 351-0198, Japan. [3]PRESTO, Japan Science and Technology (JST), Kawaguchi, Saitama 332-0012, Japan. [4]Department of Physics, Kyoto University, Kyoto 606-8502, Japan. ✉e-mail: tomotaka.kuwahara@riken.jp

effective light cone may no longer be linear[17]. Nevertheless, if the interaction decays polynomially with distance, it indicates the existence of a non-trivial effective light cone that depends on the decay rate of the interaction strength. The Lieb-Robinson bound for long-range interacting systems has been unraveled significantly in the past decade[18–24].

Conversely, the influence of the breakdown of the condition (b) has still been elusive. Considering the Lieb-Robinson velocity is roughly proportional to the interaction strength[10,25], we can no longer obtain any meaningful effective light cone without condition (b). Unfortunately, such quantum systems typically appear in quantum many-body physics because the representative examples include quantum boson Hamiltonians, which describe the atomic, molecular, and optical systems. In the absence of boson-boson interactions, one can derive the Lieb-Robinson bound with a linear light cone[26,27]. In contrast, boson-boson interactions exponentially accelerate transmitting information signals[28]. In quantum boson systems on a lattice, an arbitrary number of bosons can gather at one location, and the on-site energy can become arbitrarily large, resulting in unlimited Lieb-Robinson velocity. However, to date, there is no general established method to avoid the unboundedness of the local energy. When analyzing the systems, we must truncate the boson number at each site up to a finite number. Although practical simulations often adopt this heuristic prescription, the obtained results are always associated with some uncontrolled uncertainty. Therefore, the most pressing question is what can happen if we consider the dynamics in unconditional ways. The elucidation is crucial in the digital quantum simulation of boson systems with an efficiency guarantee.

As mentioned above, general boson systems inherently cause information propagation with unlimited speed, forcing one to restrict themselves to specific classes of interacting boson systems. The most important class is the Bose-Hubbard model, which is a minimal model comprising essential physics for cold atoms in optical lattices (see Refs. [29–31] for other boson models). In recent studies, cold atom setups have attracted significant attention as a promising platform for programmable quantum simulators[32–36]. Thus far, various researchers have explored this model in theoretical[37–42] and experimental ways[2,43,44]. Considering the Lieb-Robinson bound in the Bose-Hubbard type model, we must treat the following primary targets separately: i) transport of boson particles[45,46] and ii) information propagation[47–50]. The former characterizes the migration speed of boson particles, whereas the latter captures the propagation of all information. Relevant to the first issue i), Schuch, Harrison, Osborne, and Eisert brought the first breakthrough[45] by considering the diffusion of the initially concentrated bosons in the vacuum and ensured that the bosons have a finite propagation speed. The generalization of the result has been a challenging problem for over a decade. Recently, the initial setup has been relaxed to general states while assuming a macroscopic number of boson transport[46]. On the second issue ii), Ref. [47] derived the Lieb-Robinson velocity that was proportional to the square root of the total number of bosons. Therefore, the result provides a qualitatively better bound, whereas the velocity is still infinitely large in the thermodynamic limit. Assuming the initial state is steady and has a small number of bosons in each site, it has been proved that the effective light cone is linear with time[48,49]. Although these studies have advanced the understanding of the speed limit of Bose-Hubbard-type models, the results' application ranges are limited to specific setups, such as the steady initial state (see Ref. [50] for another example). Until now, we are far from the long-sought goal of characterizing the optimal forms of the effective light cones for the speed of i) and ii) under the condition that arbitrary time-dependent tunings of the Hamiltonian are allowed.

In this article, we overcome various difficulties and solve the problem in general setups. We treat arbitrary time-dependent Bose-Hubbard-type Hamiltonians in arbitrary dimensions starting from a

non-steady initial state. Such a setup is most natural in physics and crucial in estimating the gate complexity of digital quantum simulation of interacting boson systems. Figure 1 summarizes the main results, providing qualitatively optimal effective light cones for both the transport of boson particles and information propagation. As a critical difference between bosons and fermions (or spin models), we have clarified that the acceleration of information propagation can occur in high dimensions. Furthermore, as a practical application, we develop a gate complexity for efficiency-guaranteed digital quantum simulations of interacting bosons based on the Haah-Hastings-Kothari-Low (HHKL) algorithm[14].

## Results
### Speed limit on boson transport

We consider a quantum system on a $D$-dimensional lattice (graph) with $\Lambda$ set for all sites. For an arbitrary subset $X \subseteq \Lambda$, we denote the number of sites in $X$ by $|X|$, that is, the system size is expressed as $|\Lambda|$. We define $b_i$ and $b_i^\dagger$ as the bosonic annihilation and creation operators at the site $i \in \Lambda$, respectively. We focus on the Bose-Hubbard type Hamiltonian in the form of

$$H = \sum_{\langle i,j \rangle} J_{i,j}(b_i b_j^\dagger + \text{h.c.}) + V \tag{1}$$

with $|J_{i,j}| \le \bar{J}$ and $V := f(\{\hat{n}_i\}_{i\in\Lambda})$, where $\sum_{\langle i,j \rangle}$ is the summation for all pairs of the adjacent sites $\{i, j\}$ on the lattice and $f(\{\hat{n}_i\}_{i\in\Lambda})$ is an appropriate function of the boson number operators $\{\hat{n}_i\}_{i\in\Lambda}$ with $\hat{n}_i = b_i^\dagger b_i$. The constraints on the function $f(\{\hat{n}_i\}_{i\in\Lambda})$ depend on the specific problems under consideration. These constraints are explicitly detailed in the statements of our main Results 1–3. In Result 1, there are no restrictions on $f(\{\hat{n}_i\}_{i\in\Lambda})$; in other words, arbitrary long-range boson-boson couplings are allowed. Result 2 requires a finite interaction length, but no additional constraints. In Result 3, alongside a finite interaction length, we assume that the form of the function is polynomial. Furthermore, similar to the Lieb-Robinson bound in spin/fermion systems, all our results are applicable to Hamiltonians with arbitrary time dependences. Although any time-dependences are allowed for Results 1 and 2, we need an additional condition on the norm of the derivative as in Ineq. (36) for the proof of Result 3.

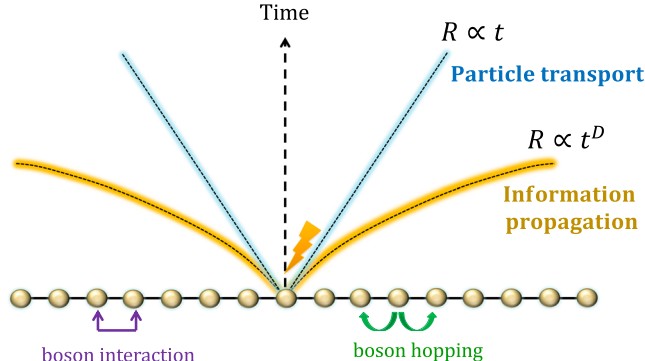

**Fig. 1 | Illustration of the effective light cones.** Herein, we describe the interacting bosons by the Bose-Hubbard type Hamiltonian (1). We first consider how fast boson particles move to distant regions, as shown in Fig. 2. The light cone for the boson particle transport is proved to be almost linear up to logarithmic corrections (denoted by the blue shaded line), as shown in Result 1. Conversely, if we consider the propagation of the full information (see also Fig. 3), the speed can be much faster than the particle transport. The effective light cone is proved to be polynomial with time, and the exponent is equal to the space dimension $D$ (denoted by the orange shaded line), where the mathematical form of the Lieb-Robinson bound is given in Result 2. We can explicitly construct a protocol to achieve the light cone using dynamics with time-dependent Bose-Hubbard type Hamiltonians (see Fig. 4).

We denote the subset Hamiltonian supported on $X \subset \Lambda$ by $H_X$, which picks up all the interactions included in $X$. The time evolution of an operator $O$ by the Hamiltonian $H_X$ is expressed as $O(H_X, t) := e^{iH_X t} O e^{-iH_X t}$. In particular, we denote $O(H, t)$ by $O(t)$ for simplicity.

We first focus on how fast the bosons spread from a region $X \subset \Lambda$ to the outside (see also Fig. 2) by adopting the notation of the extended subset $X[r]$ by length $r$ as

$$X[r] := \{i \in \Lambda | d_{i,X} \le r\}, \tag{2}$$

where $d_{i,X}$ is the distance between the subset $X$ and the site $i$. When $X$ is given by one site (i.e., $X = \{i\}$), $i[r]$ simply denotes the ball region centered at the site $i$. We here consider the time evolution of the boson number operator $\hat{n}_X := \sum_{i \in X} \hat{n}_i$. We prove that the higher order moment for boson number operator $[\hat{n}_X(t)]^s$ is upper-bounded by $\hat{n}_{X[R]}^s$ with an exponentially decaying error with $R$, i.e., $[\hat{n}_X(t)]^s \preceq \hat{n}_{X[R]}^s + e^{-\Omega(R/t)}$. Throughout this study, we use the notation $O_1 \preceq O_2$ that means $\mathrm{tr}[\sigma(O_2 - O_1)] \ge 0$ for an arbitrary quantum state $\sigma$. Our first result is roughly described by the following statement:

**Result 1.** Let us consider arbitrary boson-boson interactions $V$ in Eq. (1) without any assumptions on the function $f$. For $R \ge c_0 t \log t$, the time-evolution $\hat{n}_X(t)$ satisfies the operator inequality of

$$[\hat{n}_X(t)]^s \preceq \left[ \hat{n}_{X[R]} + \delta \hat{n}_{X[R]} + c_2 t s \right]^s,$$

where $\delta \hat{n}_{X[R]} = e^{-c_1 R/t} \sum_{j \in \Lambda} e^{-c_1' d_{j,X[R]}} \hat{n}_j$ and $\{c_0, c_1, c_1', c_2\}$ are the constants of $\mathcal{O}(1)$. The operator $\delta \hat{n}_{X[R]}$ is as small as $e^{-\Omega(R/t)}$ if there are not many bosons around the region $X[R]$. We can apply this theorem to a wide range of setups. Interestingly, it holds for systems with arbitrary long-range boson-boson interactions, such as the Coulomb interaction. Moreover, we can also apply the theorem for imaginary time evolution $\hat{n}_X(it) = e^{-tH} \hat{n}_X e^{tH}$.

From the theorem, we can see that the speed of the boson transport from one region to another is almost constant; at most, it grows logarithmically with time. This theorem gives a complete generalization of the result in Ref. 45, which discusses the boson transport for initial states that all the bosons are concentrated in a particular region. If an initial state has a finite number of bosons at each site, the probability distribution for the number of bosons still decays exponentially after a time evolution. Herein, the decay form is determined by Result 1. The estimation provides critical information for simulating the quantum boson systems with guaranteed precision (see Result 3).

Lastly, we notice that Result 1 does not imply that the operator $\hat{n}_X(t)$ is approximated onto region $X[R]$; that is, we cannot ensure $\hat{n}_X(t) \approx \hat{n}_X(H_{X[R]}, t) = e^{iH_{X[R]} t} O e^{-iH_{X[R]} t}$ as in (4). For example, if we consider a phase operator $e^{i\hat{n}_X}$, the influence can propagate acceleratingly (see below for an explicit example).

## Lieb-Robinson bound

We next consider the approximation of the time-evolved operator $O_{X_0}(t)$ by $O(H_{X_0[R]}, t)$ using the subset Hamiltonian $H_{X_0[R]}$ (see Fig. 3), where $O_{X_0}$ is an arbitrary operator. Regarding the Holevo capacity, the approximation error characterizes all the information that propagates outside the region $X_0[R]$[25]. In the following, as a natural setup, we consider an arbitrary initial state with low-boson density condition:

$$\mathrm{tr}(\rho_0 \hat{n}_i^s) \le \frac{1}{e} \left( \frac{b_0}{e} s^\kappa \right)^s, \tag{3}$$

where $b_0$ and $\kappa$ ($\ge 1$) are the constants of $\mathcal{O}(1)$. From this condition, the boson number distribution at each site decays (sub-)exponentially at the initial time. The simplest example is the Mott state, where a finite fixed number of bosons sit on each site. We emphasize that without assuming any conditions for the boson number, there is no speed limit for general information propagation. More precisely, the speed of information propagation is directly proportional to the number of bosons at local sites[41]. This underscores why the low-boson-density condition is the minimal assumption required to establish a meaningful Lieb-Robinson bound. This point also makes a clear difference between the information propagation and the particle transport, in which no conditions are imposed for initial states in Result 1.

Under the condition (3), we can prove the following statement:

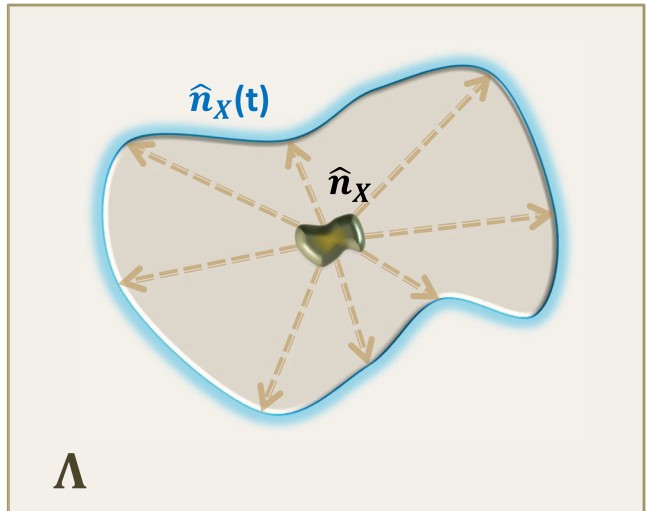

**Fig. 2 | Setup of the boson particle transport.** We consider a boson number operator $\hat{n}_X$ on a region $X$, where $\Lambda$ is the total system. After time $t$, the bosons initially concentrated on $X$ spread outside the region at a certain speed. Assuming there are no bosons outside $X$ at the initial time (i.e., $t = 0$), the boson particle transport is known to have a finite speed[45]; that is, it is approximately restricted in a region $X[R]$ with $R \approx t$ (enclosed by the blue shaded line). Result 1 generalizes the result to arbitrary initial states up to a logarithmic correction. The result plays a crucial role in truncating the boson numbers at each site after time evolution while guaranteeing the desired precision.

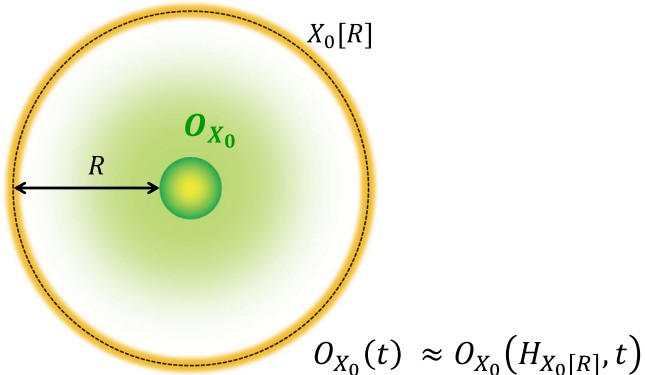

**Fig. 3 | Setup of information propagation.** We consider an operator $O_{X_0}$ supported on the subset $X_0$ and approximate the time-evolved operator $O_{X_0}(t)$ onto the extended region $X_0[R]$ (enclosed by the orange shaded line). Additionally, we assume that the boson number distribution at each site is sub-exponentially suppressed for an initial state, as in Eq. (3). Then, as long as $R \ge t^D \mathrm{polylog}(t)$, the approximation error between $O_{X_0}(t)$ by $O(H_{X_0[R]}, t)$ decays sub-exponentially with $R$. The effective light cone for the information propagation grows polynomially with time as $R \approx t^D$.

**Result 2**. Let us assume that the range of boson-boson interactions $V$ is finite in Eq. (1). Then, for an arbitrary operator $O_{X_0}$ ($\|O_{X_0}\| = 1$), the time evolution $O_{X_0}(t)$ is well approximated by using the subset Hamiltonian on $X_0[R]$ with the error of

$$\left\| \left[ O_{X_0}(t) - O_{X_0}(H_{X_0[R]}, t) \right] \rho_0 \right\|_1 \le e^{-C(R/t^D)^{\frac{1}{xD}}}, \qquad (4)$$

for $R \ge t^D \text{polylog}(t)$, where $\|\cdot\|_1$ is the trace norm and $C$ is an $\mathcal{O}(1)$ constant.

The bound (4) tells us that any operator spreading is approximated by a local operator on $X_0[R]$ with the accuracy of the right-hand side. It rigorously bounds any information propagation. A crucial observation here is that the propagation speed accelerates depending on the spatial dimensions, which is a stark difference from fermionic lattice systems.

The accelerated information propagation can be interpreted physically as follows. According to Ref. 41, the velocity of information propagation is directly proportional to the number of bosons at local sites in the presence of boson-boson interactions. Consequently, if dynamic processes cause an increase in boson concentrations within specific one-dimensional regions, the boson density in that 1D region will rise over time. As a result, information propagation on this 1D "information path" experiences acceleration. This phenomenon is specific to high-dimensional systems. In one-dimensional systems, if bosons concentrate in a specific region, the surrounding areas exhibit sparse boson densities, preventing persistent acceleration.

## Optimality of the effective light cone

As discussed in Result 2, the bound on the speed of information propagation is proportional to $t^{D-1}$. Herein, we show that the obtained upper bound is qualitatively tight by explicitly developing time evolution to achieve the bound. As the initial state, we consider the Mott state with only one boson at each site. The protocol comprises the following two steps.

1. First, we set the path of the information propagation. We transport the bosons such that they are collected on the path. Here, the path is given by the one-dimensional ladder [see Fig. 4 (a)].

2. Then, we encode the qubits on the ladder and realize the CNOT gate by using two-body interactions and boson hopping [see Fig. 4 (b)].

We here consider the first step. Let us consider two nearest neighbor sites $i_1$ and $i_2$ where the state is given by $|N\rangle_{i_1} \otimes |1\rangle_{i_2}$. We denote the Fock state on the site $i$ by $|m\rangle_i$ with $m$ the boson number. Our task is to move the bosons on the site $i_1$ to the site $i_2$, that is,

$$|N\rangle_{i_1} \otimes |1\rangle_{i_2} \rightarrow |0\rangle_{i_1} \otimes |N+1\rangle_{i_2}.$$

Such a transformation is realized by combining the free-boson Hamiltonian and Bose-Hubbard Hamiltonian, and the necessary time is inversely proportional to the hopping amplitude of the bosons $\bar{J}$ (see the Method section). Therefore, based on the time evolution of $t/2$, we can concentrate the bosons in a region within a distance of $(\bar{J}t)$ from the boson path. The boson number at one site on the boson path is now proportional to $(\bar{J}t)^{D-1}$. Herein, we denote the quantum state on the boson path by $\bigotimes_{j=1}^{\ell} |\bar{n}_t, \bar{n}_t\rangle_j$ with $\bar{n}_t \propto (\bar{J}t)^{D-1}$. Here, $\ell$ represents the total length of the information path.

In the second step, we encode the quantum states $|\bar{n}_t, \bar{n}_t\rangle_j$ and $|\bar{n}_t - 1, \bar{n}_t + 1\rangle_j$ on the $j$th row by $|1\rangle_j$ and $|0\rangle_j$, respectively, to prove that the time required to implement the controlled-NOT (CNOT) gate is at most $\mathcal{O}(t^{-D+1})$. By using appropriate two-body interactions between the $(j-1)$th row and $j$th row, no boson hopping is observed on the $j$th row when the $(j-1)$th row is given by $|\bar{n}_t, \bar{n}_t\rangle_j$; however, there exists boson hopping on the $j$th row when the $(j-1)$th row is given by $|\bar{n}_t - 1, \bar{n}_t + 1\rangle_j$. One can control the boson-boson interactions such that only the hopping between $|\bar{n}_t, \bar{n}_t\rangle_j \leftrightarrow |\bar{n}_t - 1, \bar{n}_t + 1\rangle_j$ occurs. We thus realize the CNOT operation on the information path (see the Method section), where the sufficient time required to realize it is proportional to $1/(\bar{J}\bar{n}_t) \propto 1/(\bar{J}^D t^{D-1})$. Therefore, in half of the total time $t/2$, we can implement $(\bar{J}t)^D$ pieces of the CNOT gates, which allows us to propagate the information from one site to another as long as the distance between these sites is smaller than $(\bar{J}t)^D$.

One can demonstrate that the number of CNOT operations is directly linked to the distance of the operator spread, following the discussion in Ref. 25. To illustrate this, consider two types of

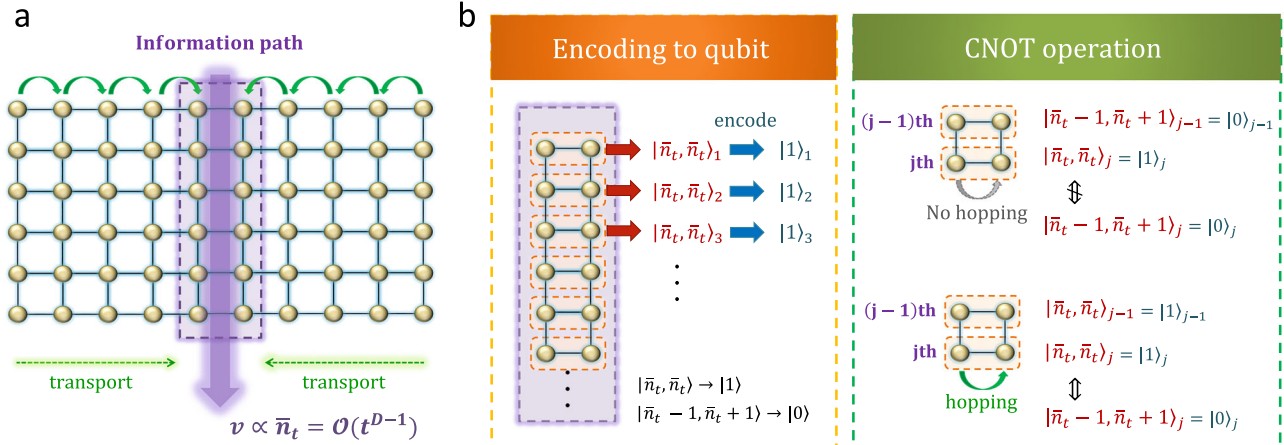

**Fig. 4 | Outline of the proposed protocol to achieve the fastest information transfer by interacting with boson systems. a** Schematic picture of information transfer. In the initial state, one boson sits at each site. In the first half, we transport the boson particles to a particular one-dimensional region called the information path. The path is now given by a ladder (denoted by the purple-shaded region). Considering the speed of the boson transport is finite, we accumulate the bosons in the region within a distance of $(\bar{J}t)$ from the path. Therefore, the number of bosons $\bar{n}_t$ at each site on the path is proportional to $(\bar{J}t)^{D-1}$ (in the picture $D = 2$). Afterward, we switch off the boson hopping and isolate the information path so that the

bosons cannot escape from the path. **b** Encoding qubits and CNOT operation. The speed of the information propagation is proportional to $\bar{n}_t$. By encoding the qubit on the $j$th row as $|\bar{n}_t, \bar{n}_t\rangle_j \rightarrow |1\rangle_j$ and $|\bar{n}_t - 1, \bar{n}_t + 1\rangle_j \rightarrow |0\rangle_j$, we can implement the CNOT operation by appropriately choosing boson-boson interactions. If the state on the $(j-1)$th row is given by $|0\rangle_{j-1}$, there is no boson hopping between the two sites on the $j$th row. In contrast, if the state on the $(j-1)$th row is given by $|1\rangle_{j-1}$, there exists a boson hopping. Herein, the hopping rate of one boson is amplified by $\bar{n}_t$ times, and hence, the transition time from $|0\rangle_j$ to $|1\rangle_j$ is proportional to $1/\bar{n}_t$. This enables us to implement the CNOT operation in the time of $1/\bar{n}_t$.

operations: flipping or non-flipping the endmost qubit on the information path at the time $t/2$, where the boson concentration has been over. For simplicty, we here prepare the quantum state on the information path so that it can be given by the product state $|0\rangle^{\otimes \ell}$ instead of $|1\rangle^{\otimes \ell}$. By flipping the endmost qubit to $|1\rangle$, $m$-sequential CNOT operations transform the state to $|1\rangle^{\otimes m}|0\rangle^{\otimes \ell-m}$, while without flipping, the state remains unchanged $|0\rangle^{\otimes \ell}$. By encoding classical information as flipping (=0) or non-flipping (=1) of the endmost qubit, one can transmit 1 bit of information through the sequence of CNOT operations. The connection between Holevo capacity and operator spreading[25] implies that this process necessarily induces the operator spreading of the flipping unitary on the endmost qubit in the Heisenberg picture. Thus, $\bar{J}^D t^D$ CNOT operations during the time $t$ achieves the Lieb-Robinson velocity of $\bar{J}^D t^{D-1}$. This accelerating information propagation must be clearly distinguished from boson transport with a constant velocity, as in Result 1.

We finally discuss a comparison with the mechanism proposed by Eisert and Gross[28]. In their model, the Hamiltonian effectively amplifies the hopping of bosons, with hopping amplitudes directly proportional to their positions. In contrast, our mechanism relies on dynamically adjusting the local boson numbers. The crucial aspect is that, in the presence of boson-boson interactions, the speed of information propagation is enhanced by the local boson numbers.

### Provably efficient digital quantum simulation

In the final application, we consider the quantum simulation of time evolution and estimate the sufficient number of quantum gates that implement the bosonic time evolution $e^{-iHt}$ acting on an initial state $\rho_0$. In detail, we prove the following statement:

**Result 3.** Let us assume that the boson-boson interactions in $V = f(\{\hat{n}_i\}_{i \in \Lambda})$ are finite in length, and the function $f(\{\hat{n}_i\}_{i \in \Lambda})$ is given by a polynomial with limited degrees and coefficients. For an arbitrary initial state $\rho_0$ with the condition (3), the number of elementary quantum gates for implementing $e^{-iHt}\rho_0 e^{iHt}$ up to an error $\epsilon$ is at most

$$|\Lambda| t^{D+1} \text{polylog}(|\Lambda| t/\epsilon), \qquad (5)$$

with the depth of the circuit $t^{D+1} \text{polylog}(|\Lambda| t/\epsilon)$, where the error is given in terms of the trace norm. Note that $|\Lambda|$ is the number of sites in the total system.

We extend the Haah-Hastings-Kothari-Low (HHKL) algorithm[14] to the interacting bosons. Before going to the algorithm, we truncate the boson number at each site up to $\bar{q}$. We define $\Pi_{\bar{q}}$ as the projection onto the eigenspace of the boson number operators $\{\hat{n}_i\}_{i \in \Lambda}$ with eigenvalues smaller than or equal to $\bar{q}$. Then, we consider the time evolution by the effective Hamiltonian $\bar{\Pi}_{\bar{q}} H \bar{\Pi}_{\bar{q}}$. In the following, for an arbitrary operator $O$, we denote the projected operator $\bar{\Pi}_{\bar{q}} O \bar{\Pi}_{\bar{q}}$ by $\tilde{O}$ for simplicity. Assuming low-boson density (3), Result 1 gives the upper bound of the approximation error of

$$\left\| \rho_0(t) - \rho_0(\tilde{H}, t) \right\|_1 \le |\Lambda| e^{-c_3 \left[ \bar{q}/(t \log t)^D \right]^{1/\kappa}}. \qquad (6)$$

Therefore, to achieve the error of $\epsilon$, we need to choose as $\bar{q} = (t \log t)^D \log^\kappa(|\Lambda|/\epsilon) = t^D \text{polylog}(|\Lambda| t/\epsilon)$.

In the algorithm, we first adopt the interaction picture of the time evolution:

$$e^{-iHt} = e^{-i\bar{V}t} \mathcal{T} e^{-i\int_0^t \tilde{H}_0(\bar{V}, x)dx}. \qquad (7)$$

First, the time evolution by $\tilde{V}$ is decomposed to $\mathcal{O}(|\Lambda|)$ pieces of the local time evolution, considering the interaction terms in $V$ commute with each other. Second, we implement the time evolution for the time-dependent Hamiltonian $\tilde{H}_0(\bar{V}, x)$. The Hamiltonian $\tilde{H}_0(\bar{V}, x)$ contains interaction terms like $e^{i\bar{V}x} \tilde{b}_i \tilde{b}_j^\dagger e^{-i\bar{V}x}$, which has a bounded norm by $\mathcal{O}(\bar{q})$ and is described by an $\mathcal{O}(1)$ sparse matrix. Herein, we say that an operator $O$ is $d$ sparse if it has at most $d$ nonzero entries in any row or column. Moreover, the norm of the derivative $\|d(e^{i\bar{V}x} \tilde{b}_i \tilde{b}_j^\dagger e^{-i\bar{V}x})/dx\|$ is upper-bounded by $\text{poly}(\bar{q})$ owing to the assumption for $f(\{\hat{n}_i\}_{i \in \Lambda})$.

Therefore, the problem is equivalent to implementing the time evolution of the Hamiltonian with the following three properties: (i) the norms of the local interaction terms are upper-bounded by $\mathcal{O}(\bar{q})$, (ii) the interaction terms are described by $\mathcal{O}(1)$ sparse matrices, and (iii) the time derivative of the local interactions has a norm of $\text{poly}(\bar{q})$ at most. Such cases can be treated by simply combining the previous works in Refs. 14 and 51. As shown in the Method section, the total number of elementary circuits is at most $(|\Lambda| t\bar{q})\text{polylog}(|\Lambda| t\bar{q}/\epsilon)$, which reduces to (5) by applying $\bar{q} = t^D \text{polylog}(|\Lambda| t/\epsilon)$.

## Discussion

Our study clarified the qualitatively tight Lieb-Robinson light cone for systems with the Bose-Hubbard type Hamiltonian. Still, we have various improvements to implement in future works. First, the obtained bounds incorporate logarithmic corrections, but there is a possibility of their removal through a refinement of the current analyses. Currently, it does not seem to be a straightforward problem to remove them using our existing techniques. A possible starting point to achieve this is to consider the difference between average values as $\text{tr}[(O_{X_0}(t) - O_{X_0}(H_{X_0[R]}, t))\rho_0]$ instead of the trace norm $\|(O_{X_0}(t) - O_{X_0}(H_{X_0[R]}, t))\rho_0\|_1$. While this quantity may not capture the propagation of total information, a significantly stronger bound can be proven in one-dimensional systems, where the light cone form strictly follows a linear form with time[49]. Through the refinement and combination of existing techniques, there is a possibility of eliminating the logarithmic corrections in our current bound in future studies. Second, although acceleration of the information propagation is possible, there are particular cases where the linear light cone is rigorously proved[48,49]. Therefore, by appropriately avoiding the acceleration mechanism depicted in our protocol, we can possibly establish a broader class that retains the linear light cone. Third, it is an interesting open question to seek the possibility of improving the current gate complexity $|\Lambda| t^{D+1} \text{polylog}(|\Lambda| t/\epsilon)$ and clarify the optimal gate complexity to simulate the quantum dynamics. We hope a more simplified technique may appear for implementing these improvements, considering the current proof techniques are rather complicated.

Other directions include generalizing the current results beyond the Bose-Hubbard type Hamiltonian (1). As a straightforward extension, it is intriguing to investigate under what conditions boson-boson interactions like $b_{i_1} b_{i_2} b_{i_3}^\dagger b_{i_4}^\dagger$ can lead to information propagation with a limited speed. In this scenario, relying solely on the low-boson-density condition proves insufficient for regulating the speed of information propagation (cf. Ref. 52). Still, such an extension has practical importance. For example, when applying the quasi-adiabatic continuation technique[7] to the Bose-Hubbard type models, we must derive the Lieb-Robinson bound for the quasi-adiabatic continuation operator, which is no longer given by the form of Eq. (1). We expect that our newer techniques will be helpful in developing interacting boson systems where the effective light cone is at most polynomial with time.

Finally, it is intriguing to experimentally or numerically observe the supersonic propagation of quantum signals using a mechanism similar to that illustrated in Fig. 4. In our protocol, we employed highly artificial boson-boson interactions. Then, a significant open problem is whether acceleration can occur even under time-independent

Hamiltonians. We anticipate that the boson transport in the initial step can be achieved by reversing the time evolution (see Supplementary Note 11). As for the second step in the protocol, Ref. 41 has already noted that the group velocity of the propagation front of correlations is proportional to the boson number at each site. Hence, we believe that the proposed acceleration mechanism can be realized within the current experimental setups.

## Methods

### Outline of the proof for Result 1

Throughout the proof, we denote $\mathcal{O}(1)$ as an arbitrary finite combination of the fundamental parameters, which are detailed in Supplementary Table 1.

First, we present several key ideas to prove Result 1 by considering $[\hat{n}_X(t)]^s$ with $s = 1$ for simplicity. By refining the technique in Ref. 45, we begin with the following statement (Supplementary Subtheorem 1):

$$\hat{n}_X(\tau) \preceq \hat{n}_X + c_{\tau,1}\hat{\mathcal{D}}_X + c_{\tau,2}, \tag{8}$$

with $\hat{\mathcal{D}}_X := \sum_{i \in \partial X}\sum_{j \in \Lambda} e^{-d_{i,j}}\hat{n}_j$, where $c_{\tau,1}$ and $c_{\tau,2}$ are the constants that grow exponentially with $\tau$, that is, $c_{\tau,1}, c_{\tau,2} = e^{\mathcal{O}(\tau)}$. We define $\partial X$ as the surface region of the subset $X$, that is, $\partial X = \{i \in X | d_{i,X^c} = 1\}$. From the definition, the operator $\hat{\mathcal{D}}_X$ is roughly given by the boson number operator around the surface region of $X$ with an exponential tail (see Fig. 5). In the inequality (8), the coefficients $c_{\tau,1}, c_{\tau,2}$ grow exponentially with $\tau$, and hence the inequality becomes meaningless for large $\tau$, which has been the main bottleneck in[45].

The key technique to overcome the above-mentioned difficulty is the connection of the short-time evolution $e^{-iH\tau}$ with $\tau$ a constant of $\mathcal{O}(1)$, which has played an important role in the previous works[22,48,53]. We first refine the upper bound (8) to

$$\hat{n}_X(\tau) \preceq \hat{n}_{X[\ell]} + e^{-\Omega(\ell)} + c_{\tau,2}. \tag{9}$$

If the above inequality holds, by iteratively connecting the short-time evolution $(t/\tau)$ times, we obtain

$$\hat{n}_X(t) \preceq \hat{n}_{X[(t/\tau)\ell]} + \frac{t}{\tau}\left(e^{-\Omega(\ell)} + c_{\tau,2}\right), \tag{10}$$

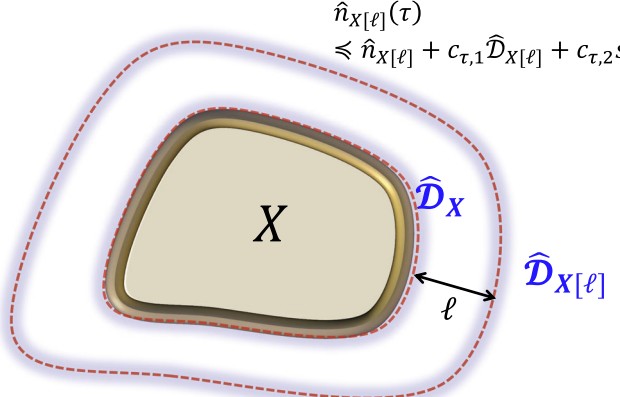

$$\hat{n}_{X[\ell]}(\tau)$$
$$\preceq \hat{n}_{X[\ell]} + c_{\tau,1}\hat{\mathcal{D}}_{X[\ell]} + c_{\tau,2}s$$

**Fig. 5 | Time evolution of boson number operator.** A short-time evolution of $\hat{n}_X(\tau)$ is bounded from above by $\hat{n}_X + c_{\tau,1}\hat{\mathcal{D}}_X + c_{\tau,2}$. The influence of $\hat{\mathcal{D}}_X$ exponentially decays with the distance from the surface region $\partial X$ (denoted by the blue shaded region). The contribution of $\hat{\mathcal{D}}_X$ may be fatal in some classes of the initial states. If all bosons concentrate on $\partial X$ in an initial state $\rho$, $\text{tr}(\rho\hat{\mathcal{D}}_X)$ can be as large as $\text{tr}(\rho\hat{n}_X)$. However, by considering the time evolution of $\hat{n}_{X[\ell]}(\tau)$, the contribution by the bosons on $\partial X$ is exponentially small with $\ell$ in the operator $\hat{\mathcal{D}}_{X[\ell]}$. This motivates us to consider the minimization problem (12) for all choices of $X[r]$ ($0 \leq r \leq \ell$) to derive the upper bound (13).

which yields the desired inequality in Result 1 for $s = 1$ by choosing $R = (t/\tau)\ell$ (or $\ell = \tau R/t$).

Then, we aim to derive the bound (9) using the inequality (8). However, the derivation is not straightforward because the inequality $\hat{n}_X + c_{\tau,1}\hat{\mathcal{D}}_X + c_{\tau,2} \preceq \hat{n}_{X[\ell]} + e^{-\Omega(\ell)} + c_{\tau,2}$ does not hold in general. For example, let us consider a quantum state $\rho$ such that all bosons concentrate on $\partial X$ (see Fig. 5). Then, we have

$$\text{tr}(\rho\hat{\mathcal{D}}_X) \propto \text{tr}(\rho\hat{n}_X), \tag{11}$$

which makes $\text{tr}[\rho(\hat{n}_X + c_{\tau,1}\hat{\mathcal{D}}_X)] = [1 + \Omega(1)]\text{tr}(\rho\hat{n}_X)$. Therefore, connecting the time evolution $(t/\tau)$ times yields an exponential term $[1 + \Omega(1)]^{t/\tau}$. To avoid such exponential growth, we first upper-bound $\hat{n}_X(\tau) \preceq \hat{n}_{X[\ell]}(\tau)$ for $\forall \tau > 0$ as a trivial bound, where $\ell$ ($\geq 0$) can be arbitrarily chosen. Then, we use the inequality (8) to obtain

$$\hat{n}_X(\tau) \preceq \hat{n}_{X[\ell]}(\tau) \preceq \hat{n}_{X[\ell]} + c_{\tau,1}\hat{\mathcal{D}}_{X[\ell]} + c_{\tau,2}.$$

The point here is that $\hat{\mathcal{D}}_{X[\ell]}$ is localized around the surface of $X[\ell]$ with an exponentially decaying tail. Using the above operator inequality, we can resolve the drawback in Eq. (11) that originates from the concentration around the boundary $\partial X$. Here, for the quantum state $\rho$ which has the boson concentration on the region $\partial X$, we have

$$\text{tr}(\rho\hat{\mathcal{D}}_{X[\ell]}) \approx e^{-\Omega(\ell)}\text{tr}(\rho\hat{n}_X)$$

instead of Eq. (11). Therefore, the contribution from the operator $\hat{\mathcal{D}}_{X[\ell]}$ is exponentially small with the length $\ell$.

From the above discussion, to derive a meaningful upper bound for short-time evolution, we need to consider

$$\min_{r:0 \leq r \leq \ell} \text{tr}\left[\rho\left(\hat{n}_{X[r]} + c_{\tau,1}\hat{\mathcal{D}}_{X[r]} + c_{\tau,2}\right)\right] \tag{12}$$

for an arbitrary quantum state $\rho$, which also gives an upper bound of $\text{tr}[\rho\hat{n}_X(\tau)]$. We cannot solve the optimization problem (12) in general but ensure the existence of $r \in [0, \ell]$ that satisfies the following inequality (see Supplementary Lemma 13 and Supplementary Proposition 16):

$$\hat{n}_X(\tau) \preceq \hat{n}_{X[\ell]} + c_{\tau,1}\delta_\ell(\hat{n}_{X[\ell]} + \hat{\mathscr{D}}_{X[\ell]}) + c_{\tau,2}, \tag{13}$$

where $\delta_\ell$ decays exponentially with $\ell$, i.e., $\delta_\ell = e^{-\Omega(\ell)}$, and we define $\hat{\mathscr{D}}_{X[\ell]} = \sum_{j \in X[\ell]^c} e^{-3d_{j,X[\ell]}/4}\hat{n}_j$. The inequality (13) is given in the form of the desired inequality (9), which also yields the upper bound (10) (Supplementary Theorem 1). More precisely, the iterative use of the inequality (13) yields an additional coefficient $(1 + c_{\tau,1}\delta_\ell)^{t/\tau}$ to the first term $\hat{n}_{X[(t/\tau)\ell]}$ in (10). We need the condition $R \geq c_0 t \log t$ (or $\ell \propto \log(t)$) to ensure $(1 + c_{\tau,1}\delta_\ell)^{t/\tau} \lesssim 1 + c_{\tau,1}t\delta_\ell$. Therefore, we prove the main inequality in Result 1.

For a general $s$th moment, we apply similar analyses to the case of $s = 1$. As a remark, we cannot simply obtain $[\hat{n}_X(\tau)]^s \preceq (\hat{n}_X + c_{\tau,1}\hat{\mathcal{D}}_X + c_{\tau,2})^s$ from the inequality (8) because $O_1 \preceq O_2$ does not imply $O_1^s \preceq O_2^s$ in general. Instead, we obtained the following modified upper bound:

$$[\hat{n}_X(\tau)]^s \preceq (\hat{n}_X + c_{\tau,1}\hat{\mathcal{D}}_X + c_{\tau,2}s)^s. \tag{14}$$

Then, we consider a similar procedure to the optimization problem (12) and obtain an analogous inequality to (13) (Supplementary Proposition 18). This allows us to connect the short-time evolution to derive Result 1 for general $s$.

## Non-acceleration of Boson transport

Here, we demonstrate that the protocol in Fig. 4 cannot induce the acceleration of boson transport. While this protocol enables the transformation

$$(b_i^\dagger)^m |M_1\rangle \to (b_j^\dagger)^m |M_1\rangle \qquad (15)$$

as long as $d_{i,j} \lesssim t^D$ (where $|M_1\rangle$ is the Mott state with one boson at each site), this process does not imply genuine particle transport due to the indistinguishability of bosons. In the first place, even without the Fig. 4 protocol, the transformation (15) can be achieved in a constant time for arbitrary distances (see Fig. 6). To characterize particle transport, it is essential to ensure that the increased bosons indeed originate from a distant region. This can be achieved in the following cases:

1. If $\mathrm{tr}(\rho(t)\hat{n}_X) > \mathrm{tr}(\rho\hat{n}_{X[R]})$, we can ensure that a part of the increase in boson number comes from the region $X[R]^c$, achieving particle transport over a distance $R$.
2. By making target bosons distinguishable from others (e.g., bosons with the spin degree of freedom), particle transport can be clearly defined. The first case is addressed in Result 1, where we establish a finite speed. In the second case, we also prove the finite speed of transport by slightly generalizing Result 1. In this case, the Hamiltonian should be generalized to

$$H = \sum_\sigma \sum_{\langle i,j \rangle} J_{i,j}(b_{i,\sigma} b_{j,\sigma}^\dagger + \mathrm{h.c.}) + f\left(\{\hat{n}_{i,\sigma}\}_{i\in\Lambda,\sigma}\right). \qquad$$

Then, the same operator inequality as in Result 1 holds for $\hat{n}_{X,\sigma}(t)$ for corresponding spin degrees $\sigma$.

In the context of this discussion, a more phenomenological explanation to ensure the finite velocity of boson transport is through the particle current. The particle current operator $\hat{\mathcal{J}}_{i,i+1}$ between the sites $i$ and $i+1$ is defined as

$$\hat{\mathcal{J}}_{i,i+1} := J(ib_i b_{i+1}^\dagger + \mathrm{h.c.}), \qquad (16)$$

where we consider a one-dimensional system for simplicity, and the free Hamiltonian is $H_0 = \sum_i J(b_i b_{i+1}^\dagger + \mathrm{h.c.})$. Usually, the current is defined as the product of particle density and velocity, giving the speed of particle velocity as

$$v_{\mathrm{transport}} \sim \left\| \frac{\hat{\mathcal{J}}_{i,i+1}}{\hat{n}_i + \hat{n}_{i+1}} \right\| \le 2J, \qquad (17)$$

where we use the operator inequality of $|\hat{\mathcal{J}}_{i,i+1}| \preceq 2J(\hat{n}_i + \hat{n}_{i+1})$ from $|b_i b_{i+1}^\dagger| \preceq \hat{n}_i + \hat{n}_{i+1}$ (see Supplementary Equation 459). Although it is non-trivial to derive our Result 1 only from this discussion, it provides a simple picture of why the speed of particle transport has a finite speed.

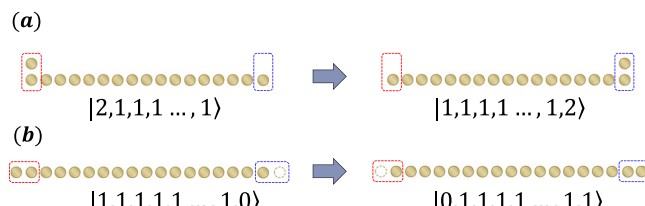

(a)

$|2,1,1,1 \dots, 1\rangle$ $|1,1,1,1 \dots, 1,2\rangle$

(b)

$|1,1,1,1,1 \dots, 1,0\rangle$ $|0,1,1,1,1 \dots, 1,1\rangle$

**Fig. 6 | Transformation from $b_1^\dagger|M_1\rangle$ to $b_n^\dagger|M_1\rangle$ (a).** This process is equivalent to one hopping from left to right of all bosons (**b**), where the left-end two sites and the right-end two sites are merged into one site, respectively. Then, this process takes time of $\mathcal{O}(1)$ for arbitrarily long 1D chains.

## Outline of the proof for Result 2: simpler but looser bound

Herein, we show how to derive the Lieb-Robinson bound with the effective light cone of $R \propto t^D$ for information propagation. The number of bosons created by the operator $O_{X_0}$, say $q_0$, is assumed to be an $\mathcal{O}(1)$ constant for simplicity. In Supplementary Notes 7, 8, 9, and 10, we treat generic $q_0$, and the obtained Lieb-Robinson bounds depend on $q_0$ (see Supplementary Theorems 2, 3, and 4). Before going to the tight Lieb-Robinson bound, we show the derivation of a looser light cone by using the truncation of the boson number as in Ref. 48, which gives $R \propto t^{D+1}$ (Supplementary Theorem 2). By applying Result 1 with $X = \{i\}$, the time evolution of the boson number operator $\hat{n}_i$ is roughly upper-bounded by the boson number on the ball region $i[\ell_t]$, that is, $\hat{n}_{i[\ell_t]}$, where $\ell_t = \mathcal{O}(t \log t)$, and we ignored the non-leading terms. Therefore, if an initial state $\rho_0$ has a finite number of bosons at each site, the upper bound of the $s$th moment after a time evolution can be given as

$$\mathrm{tr}\left[\rho_0(t)\hat{n}_i^s\right] \lesssim \mathrm{tr}\left[\rho_0 \hat{n}_{i[\ell_t]}^s\right] \propto (\ell_t^D s^\kappa)^s, \qquad (18)$$

where we use the condition (3) in the second inequality. The above inequality characterizes the boson concentration by the time evolution to ensure that the probability distribution of the boson number decays subexponentially

$$\mathrm{tr}\left[\rho_0(t)\Pi_{i,\ge x}\right] \lesssim e^{-(x/\ell_t^D)^{1/\kappa}}, \qquad (19)$$

where $\Pi_{i,\ge x}$ is the projection onto the eigenspace of $\hat{n}_i$ with the eigenvalues larger than or equal to $x$. Therefore, we expect that the boson number at each site can be truncated up to $\mathcal{O}(\ell_t^D) = t^D \mathrm{polylog}(t)$ with guaranteed efficiency.

When deriving the Lieb-Robinson bound, we adopt the projection $\bar{\Pi}_{L,\bar{q}}$ ($L \subseteq \Lambda$) such that

$$\bar{\Pi}_{L,\bar{q}} := \prod_{i\in L} \Pi_{i,\le\bar{q}}. \qquad (20)$$

This truncates the boson number at each site in the region $L$ up to $\bar{q}$. Therefore, the Hamiltonian $\bar{\Pi}_{L,\bar{q}} H \bar{\Pi}_{L,\bar{q}}$ has a finitely bounded energy in the region $L$ under the projection. The problem is whether we can approximate the exact dynamics $e^{-iHt}$ by using the effective Hamiltonian as $e^{-i\bar{\Pi}_{L,\bar{q}} H \bar{\Pi}_{L,\bar{q}} t}$. Generally, the error between them is not upper-bounded unless we impose some restrictions on the initial state $\rho_0$. Under the condition (3) of the low-boson density, the inequality (19) indicates that the dynamics may be well-approximated by $\bar{\Pi}_{L,\bar{q}} H \bar{\Pi}_{L,\bar{q}}$ as long as $\bar{q} \gg \ell_t^D$. Indeed, we can prove the following error bound similar to (6) (Supplementary Proposition 30):

$$\left\| \left( O_{X_0}(t) - O_{X_0}(\bar{\Pi}_{L,\bar{q}} H \bar{\Pi}_{L,\bar{q}}, t) \right)\rho_0 \right\|_1 \le |L| e^{-c_3 \left[\bar{q}/(t\log t)^D\right]^{1/\kappa}}. \qquad (21)$$

Following the analyses in Ref. 48, we only have to truncate the boson number in the region $X_0[R]$, that is, $L = X_0[R]$, to estimate the error $\|(O_{X_0}(t) - O_{X_0}(H_{X[R]}, t))\rho_0\|_1$.

For the effective Hamiltonian $\bar{\Pi}_{L,\bar{q}} H \bar{\Pi}_{L,\bar{q}}$, the Lieb-Robinson velocity is proportional to $\bar{q}$, and hence, if $\bar{q}t \lesssim R$, we can ensure that the time-evolved operator $O_{X_0}(\bar{\Pi}_{L,\bar{q}} H \bar{\Pi}_{L,\bar{q}}, t)$ is well-approximated in the region $X_0[R]$ (Supplementary Lemma 35). Therefore, by choosing $\bar{q} \propto R/t$ (see Supplementary Equation 527 for the explicit choice) in (21), the Lieb-Robinson bound is derived as follows:

$$\left\| \left( O_{X_0}(t) - O_{X_0}(H_{X[R]}, t) \right)\rho_0 \right\|_1 \le \exp\left[ -c\left(\frac{R}{t(t\log t)^D}\right)^{1/\kappa} + \log(|X_0[R]|) \right]. \qquad (22)$$

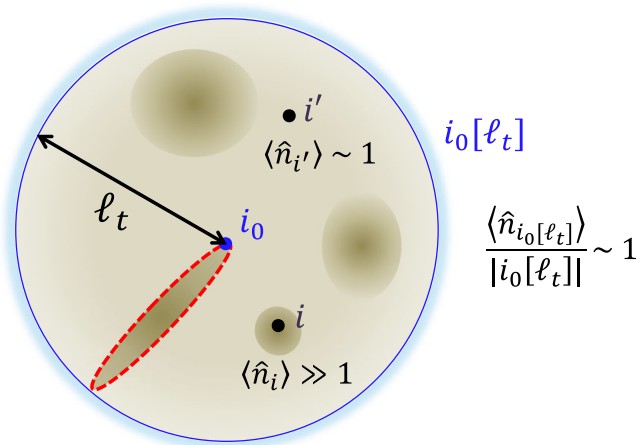

**Fig. 7 | Average of the boson number on a region $i_0[\ell_t]$ with $\ell_t = \mathcal{O}(t \log(t))$.** Even if bosons can concentrate on a few sites, the average number of bosons on one site is upper-bounded by a constant, as in (23). The local energy associated with one site is roughly proportional to the boson number on the site; hence, the average of the local energy is finitely bounded. However, if bosons concentrate onto a one-dimensional region (enclosed by the red dashed line), they induce an acceleration of information propagation (also see Fig. 4).

This gives the effective light cone in the form of $R = t^{D+1} \text{polylog}(t)$.

### Outline of the proof for Result 2: optimal light cone

To refine the bound (22), we must utilize the fact that the boson number at each site cannot be as large as $\mathcal{O}(t^D)$ simultaneously (see Fig. 7). From Result 1, after time evolution, the boson number operator $\hat{n}_{i_0[\ell_t]}$ in the ball region $i_0[\ell_t]$ is roughly upper-bounded by that in the extended ball region $i_0[2\ell_t]$ with $\ell_t = \mathcal{O}(t \log t)$. We thus obtain

$$\frac{\hat{n}_{i_0[\ell_t]}(t)}{|i_0[\ell_t]|} \lesssim \frac{\hat{n}_{i_0[2\ell_t]}}{|i_0[\ell_t]|} = \frac{\hat{n}_{i_0[2\ell_t]}}{|i_0[2\ell_t]|} \cdot \frac{|i_0[2\ell_t]|}{|i_0[\ell_t]|}. \tag{23}$$

Considering $|i_0[2\ell_t]|/|i_0[\ell_t]|$ is upper-bounded by the $\mathcal{O}(1)$ constant, the average boson number in the region $i_0[\ell_t]$ is still constant as long as the initial state satisfies $\langle \hat{n}_{i_0[2\ell_t]} \rangle / |i_0[2\ell_t]| = \mathcal{O}(1)$. We can ensure that the average local energy is upper-bounded by a constant value from the upper bound on the average number of bosons. This inspires a feeling of hope to derive a constant Lieb-Robinson velocity. Unfortunately, such an intuition does not hold, considering bosons clump together to make an information path with high boson density, as shown in Fig. 4. In such a path, up to $\mathcal{O}(\ell_t^{D-1})$, bosons sit on the sites simultaneously. Therefore, our task is to prove that the fastest information propagation occurs when bosons clump onto a one-dimensional region.

To address the aforementioned point, we need to consider cases where the interaction strengths in a Hamiltonian depend on the locations. In the standard Lieb-Robinson bound, the Lieb-Robinson velocity is proportional to the maximum local energy[9,10]. However, this estimation is insufficient when deriving the bosonic Lieb-Robinson bound, as the local energy depends on the boson number at the local site and can be as large as $\mathcal{O}(t^D)$; our current goal is to derive the Lieb-Robinson velocity as $t^{D-1}$.

For this purpose, we consider a general Hamiltonian in the form of $H = \sum_{Z \subset \Lambda} h_Z$ with the additional constraint:

$$\frac{1}{m} \sum_{j=1}^{m} \left\| h_{Z_j} \right\| \le \frac{\bar{g}_0}{m} + \bar{g}_1 \quad (\forall m \in \mathbb{N}), \tag{24}$$

where $\{h_{Z_j}\}_{j=1}^{m}$ are arbitrary interaction terms acting on subsets $\{Z_j\}_{j=1}^{m}$, respectively. Roughly speaking, the parameter $\bar{g}_0$ corresponds to the

maximum local energy on one site, and $\bar{g}_1$ is the average local energy on one site. Under the above condition, each interaction term $h_Z$ has an upper bound of $\bar{g}_0 + \bar{g}_1$. Thus, the standard Lieb-Robinson bound gives a Lieb-Robinson velocity of $\mathcal{O}(\bar{g}_0 + \bar{g}_1)$, which can be unfavorable if $\bar{g}_0$ is large. Through refined analyses, we can prove that the improved Lieb-Robinson velocity depends on the distance as $\mathcal{O}(\bar{g}_0/R) + \mathcal{O}(\bar{g}_1)$, eventually becoming $\mathcal{O}(\bar{g}_1)$ for sufficiently large $R$ (Supplementary Lemma 42). To apply this technique to the boson systems, we may come up with an idea to perform site-dependent boson number truncation instead of the uniform truncation $\bar{\Pi}_{L,\bar{q}}$ in Eq. (20). For example, we consider a projection as

$$\bar{\Pi}_{L,\mathbf{q}} := \prod_{i \in L} \Pi_{i, \le q_i}, \quad \mathbf{q} := \{q_i\}_{i \in L}. \tag{25}$$

By the projection, the effective Hamiltonian $\bar{\Pi}_{L,\mathbf{q}} H \bar{\Pi}_{L,\mathbf{q}}$ satisfies a similar condition to (24).

The primary challenge arises from the inability to obtain an accurate approximation for dynamics using the effective Hamiltonian $\bar{\Pi}_{L,\mathbf{q}} H \bar{\Pi}_{L,\mathbf{q}}$ for a specific choice of $\mathbf{q}$. This challenge is rooted in the superposition of quantum states with diverse boson configurations. For instance, consider a quantum state $|\psi\rangle$ represented as the superposition of two states, $|\psi_1\rangle$ and $|\psi_2\rangle$, where $\bar{\Pi}_{L,\mathbf{q}_1}|\psi_1\rangle = |\psi_1\rangle$ and $\bar{\Pi}_{L,\mathbf{q}_2}|\psi_2\rangle = |\psi_2\rangle$. Then, time evolution with the effective Hamiltonian $\bar{\Pi}_{L,\mathbf{q}_1} H \bar{\Pi}_{L,\mathbf{q}_1}$ provides a reliable approximation for $e^{-iHt}|\psi_1\rangle$ but not for $e^{-iHt}|\psi_2\rangle$. Conversely, time evolution with $\bar{\Pi}_{L,\mathbf{q}_2} H \bar{\Pi}_{L,\mathbf{q}_2}$ gives a good approximation for $e^{-iHt}|\psi_2\rangle$ but not for $e^{-iHt}|\psi_1\rangle$. Thus, a specific boson number truncation using $\bar{\Pi}_{L,\mathbf{q}}$ cannot be applied uniformly to all superposed states. It is necessary to consider different boson number truncations depending on the boson configuration of the superposed states, as discussed in Supplementary Note 8A.

To resolve the problem, we utilize the connection of short-time unitary evolution[22,48,53]. Let $\tau$ be a unit of time that is appropriately chosen afterward. If we can obtain the approximation error of

$$\left\| \left[ O_X(\tau) - O_X(H_{X[\ell]}, \tau) \right] \rho_0(t_1) \right\|_1 \tag{26}$$

for arbitrary $O_X$ with $X \subseteq X_0[R]$ and $t_1 \le t$, we can connect the approximation to obtain the desired error bound $\| [O_{X_0}(t) - O_{X_0}(H_{X[R]}, t)] \rho_0(t) \|_1$ (see Supplementary Equation 764). For sufficiently small $\tau$, the time-evolved state approximately preserves the initial boson distribution.

To estimate the norm (26), we consider a set of projection $\{\mathcal{P}_s\}_{s=1}^{M}$ such that $\sum_{s=1}^{M} \mathcal{P}_s = 1$, each of which constraints the boson number on the sites. By using them, we upper-bound the norm (26) by

$$\sum_{s=1}^{M} \left\| \left[ O_X(\tau) - O_X(H_{X[\ell]}, \tau) \right] \mathcal{P}_s \right\| \cdot \left\| \mathcal{P}_s \rho_0(t_1) \right\|_1. \tag{27}$$

Therefore, although the state $\rho_0(t_i)$ includes various boson number configurations, we can separately treat them. Because the summation (27) increases with the number of projections $M$, we need to select a minimal set of $\{\mathcal{P}_s\}_{s=1}^{M}$ to achieve our goal. The choice of the projections is rather technical (see Supplementary Note 9B).

Now, the short-time evolution does not drastically change the original boson number distribution. Hence, we perform boson number truncation $\bar{\Pi}_{L,\mathbf{q}}$ with $q_i$ roughly determined based on the initial boson number around the site $i$.

In conclusion, we can derive the following upper bound (see Supplementary Proposition 45):

$$\left\| \left[ O_X(\tau) - O_X(H_{X[\ell]}, \tau) \right] \rho_0(t_1) \right\|_1 \lesssim e^{-(Q/\ell^D)^{1/\kappa}} + \left( \frac{\tau Q \log(q)}{\ell^2} \right)^\ell, \quad (28)$$

where $Q$ is an arbitrary control parameter. By choosing $Q$ and $\tau$ appropriately and connecting the short-time evolution, we can prove the main statement (4) (see Supplementary Equations 769, 773, and 774 of Supplementary Note 9B). As a final remark, in the case of one-dimensional systems, we cannot utilize the original unitary connection technique[22,48,53] and have to utilize a refined version (see Supplementary Note 10).

## Realization of the CNOT operation
In the protocol to achieve the information propagation in Fig. 4, we need to implement the following two operations that involve two and four sites, respectively.

$$|N,1\rangle \leftrightarrow |0,N+1\rangle, \quad (29)$$

and

$$\begin{aligned} |\bar{n}_t, \bar{n}_t\rangle \otimes |\bar{n}_t, \bar{n}_t\rangle &\leftrightarrow |\bar{n}_t, \bar{n}_t\rangle \otimes |\bar{n}_t - 1, \bar{n}_t + 1\rangle, \\ |\bar{n}_t - 1, \bar{n}_t + 1\rangle \otimes |\bar{n}_t, \bar{n}_t\rangle &\rightarrow |\bar{n}_t - 1, \bar{n}_t + 1\rangle \otimes |\bar{n}_t, \bar{n}_t\rangle, \\ |\bar{n}_t - 1, \bar{n}_t + 1\rangle \otimes |\bar{n}_t - 1, \bar{n}_t + 1\rangle & \\ &\rightarrow |\bar{n}_t - 1, \bar{n}_t + 1\rangle \otimes |\bar{n}_t - 1, \bar{n}_t + 1\rangle, \end{aligned} \quad (30)$$

where we denote the product state $|1\rangle \otimes |N\rangle$ by $|1,N\rangle$ for simplicity. We also label the four sites as 1, 2, 3 and 4.

To achieve the operation (29), we first transform $|N,1\rangle \rightarrow |1,N\rangle$, which is achieved by the free boson Hamiltonian, that is, $H_0 = J(b_1^\dagger b_2 + h.c.)$ $(J \leq \bar{J})$. Second, to transform $|1,N\rangle \rightarrow |0,N+1\rangle$, we use the Bose-Hubbard Hamiltonian as

$$H = H_0 + h\hat{n}_2 - U\hat{n}_2^2, \quad h = (2N+1)U. \quad (31)$$

By letting $V = h\hat{n}_2 - U\hat{n}_2^2$, we get $\langle 1,N|V|1,N\rangle = \langle 0,N+1|V|0,N+1\rangle = UN(N+1)$ and $\langle j, N+1-j|V|j, N+1-j\rangle \leq UN(N+1) - 2U$ for $\forall j \in [2, N+1]$. In the limit of $U \rightarrow \infty$, the time evolution $e^{-iHt}|1,N\rangle$ is described by the superposition of the two states of $|1,N\rangle$ and $|0,N+1\rangle$. Therefore, we achieve the transformation $|1,N\rangle \rightarrow |0,N+1\rangle$ within a time proportional to $J^{-1}$.

The second operation (30) is constructed by the following Hamiltonian

$$H = H_0 + h(\hat{n}_2 - \hat{n}_1)\hat{n}_3 + U(\hat{n}_3\hat{n}_4 + \hat{n}_4 - \bar{n}_t), \quad (32)$$

where we choose $h$ to be infinitely large. Owing to the term $h(\hat{n}_2 - \hat{n}_1)\hat{n}_3$ $(h \rightarrow \infty)$, the hopping between the site 3 and 4 cannot occur unless the number of bosons on sites 1 and 2 are equal. Therefore, we achieve the second and the third operations in (30). Next, we denote $V = h(\hat{n}_2 - \hat{n}_1)\hat{n}_3 + U(\hat{n}_3\hat{n}_4 + \hat{n}_4 - \bar{n}_t)$. For an arbitrary state as $|\bar{n}_t, \bar{n}_t\rangle \otimes |\bar{n}_t - j, \bar{n}_t + j\rangle$, the eigenvalue of $V$ is given by

$$\langle \bar{n}_t, \bar{n}_t, \bar{n}_t - j, \bar{n}_t + j|V|\bar{n}_t, \bar{n}_t, \bar{n}_t - j, \bar{n}_t + j\rangle = U\left(\bar{n}_t^2 - j^2 + j\right). \quad (33)$$

Then, the eigenvalue has the same value only for $j = 0$ and $j = 1$, whereas the other eigenvalues are separated from each other by a width larger than or equal to $2U$. Therefore, by letting $U \rightarrow \infty$, the first operation (30) can be realized by following the same process as described for (29).

## Gate complexity for quantum simulation
We here derive the gate complexity to simulate the time evolution by $\tilde{H}_0(\tilde{V}, x)$ in Eq. (7). The technique herein is similar to the one in Ref. 54, which analyzes the quantum simulation for the Bose-Hubbard model with a sufficiently small boson density. Under the decomposition of Eq. (7), we must consider the class of time-dependent Hamiltonians as

$$H_t = \sum_{Z \subset \Lambda} h_{t,Z}, \quad (34)$$

where each interaction term $\{h_{t,Z}\}_{Z \subset \Lambda}$ is given by the form of $e^{i\tilde{V}t} \tilde{b}_i^\dagger \tilde{b}_j e^{-i\tilde{V}t}$. Here, $h_{t,Z}$ satisfies

$$\max_{i \in \Lambda} \sum_{Z:Z \ni i} \|h_{t,Z}\| \leq g = \mathcal{O}(\bar{q}), \quad (35)$$

and

$$\left\| \frac{dh_{t,Z}}{dt} \right\| \leq g' = \text{poly}(\bar{q}), \quad (36)$$

where $\bar{q}$ is defined by the boson number truncation as in (6). Additionally, the local Hilbert space on one site has a dimension of $\bar{q} + 1$, whereas the matrix representing $h_{t,Z}$ is $d$ sparse matrix with $d = \mathcal{O}(1)$; that is, it has at most $d$ nonzero elements in any row or column.

Now, we consider the subset Hamiltonian $h_{t,L}$ on an arbitrary subset $L$, defined as

$$H_{t,L} = \sum_{Z \subset L} h_{t,Z}. \quad (37)$$

Then, we consider the gate complexity to simulate the dynamics $U_L(0 \rightarrow \tau) := \mathcal{T}e^{-i\int_0^\tau H_{t,L}dt}$, which has been thoroughly investigated[51,55]. The Hilbert space on the subset $L$ has dimensions of $(\bar{q} + 1)^{|L|}$, and hence, the number of qubits to represent the Hilbert space is given by $|L|\log_2(\bar{q} + 1)$. Additionally, $H_L(t)$ is given by an $\mathcal{O}(d_L)$ sparse matrix, where $d_L = d \times \mathcal{O}(|L|)$. Therefore, by employing Theorem 2.1 in Ref. 51, the gate complexity for simulating $U_L(0 \rightarrow \tau)$ up to an error $\epsilon$ is upper-bounded by

$$\frac{\tilde{\tau}\log(\tilde{\tau}/\epsilon)\log[(\tilde{\tau} + \tilde{\tau}')/\epsilon]}{\log\log(\tilde{\tau}/\epsilon)}|L|\log_2(\bar{q} + 1) \quad (38)$$

with $\tilde{\tau} := d_L^2 g|L|\tau$ and $\tilde{\tau}' := d_L^2 g'|L|\tau$. By using the inequalities (35) and (36) and $d_L = \mathcal{O}(|L|)$, the above quantity reduces to the form of

$$\tau\bar{q}|L|^4\log^2(\tau\bar{q}|L|/\epsilon)\log(\bar{q}). \quad (39)$$

In the following, we consider the Haah-Hastings-Kothari-Low algorithm[14] to the time evolution of the total system $\Lambda$, that is, $U_\Lambda(0 \rightarrow t)$ by splitting the total time $t$ into $m_0 := t/\Delta t$ pieces and choosing $\Delta t$ as $\mathcal{O}(1/\bar{q})$. Then, we decompose the total system into blocks $\{B_s\}_{s=1}^{\bar{n}}$, i.e., $\Lambda = \bigcup_{s=1}^{\bar{n}} B_s$, where each block has the size of $\ell$ (see Fig. 8). We then approximate

$$U_\Lambda(0 \rightarrow t) = \mathcal{T}e^{-i\int_0^t H_x dx} \approx U_1 U_2 \cdots U_{m_0}, \quad (40)$$

where $U_j$ is an approximation for the dynamics from $(j-1)\Delta t$ to $j\Delta t$ as follows:

$$U_j = \prod_{s:\text{odd}} U_{j,B_{s,s+1}} \prod_{s=2}^{\bar{n}-1} U_{j,B_s}^\dagger \prod_{s:\text{even}} U_{j,B_{s,s+1}}. \quad (41)$$

Herein, we define $B_{s,s+1} := B_s \cup B_{s+1}$ and define $U_{j,L}$ $(L \subseteq \Lambda)$ as

$$U_{j,L} := \mathcal{T}e^{-i\int_{(j-1)\Delta t}^{j\Delta t} H_{x,L}dx}. \quad (42)$$

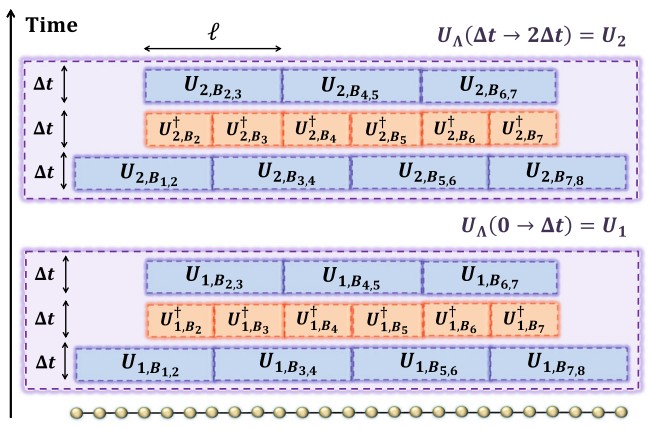

**Fig. 8 | Schematic of the HHKL decomposition (40) in a one-dimensional case.** Each unitary operator $U_{j,B_{s,s+1}}$ is given by the time evolution of the subset Hamiltonian $H_{t,B_{s,s+1}}$ from the time $(j-1)\Delta t$ to $j\Delta t$. The error of the decomposition depends on the block size, given as $e^{-\mu\ell+\nu g\Delta t}$. The local energy of the Hamiltonian $H_t$ is characterized by the constant $g = \mathcal{O}(\bar{q})$ as in Eq. (35). When estimating the gate complexity, we choose $\Delta t \propto 1/g = \mathcal{O}(1/\bar{q})$ and $\ell = \log(|\Lambda|t/\epsilon)$.

From Ref. 14, the approximation error of the decomposition (40) is given as

$$\left\| U_{\Lambda}(0 \to t) - U_1 U_2 \cdots U_{m_0} \right\| \lesssim |\Lambda| \frac{t}{\Delta t} e^{-\mu\ell + \nu g\Delta t}, \tag{43}$$

where $\mu$ and $\nu$ are the constants of $\mathcal{O}(1)$. Due to $\Delta t = \mathcal{O}(1/\bar{q})$ and $g = \mathcal{O}(\bar{q})$ from (35), we have $g\Delta t = \mathcal{O}(1)$; therefore, by choosing $\ell = \log(|\Lambda|t/\epsilon)$, we ensure that the error (43) is $< \epsilon$.

We now have all the ingredients to estimate the gate complexity. From the estimation (39), each unitary operator $U_{j,B_s}$ was implemented with a gate complexity of

$$|B_s|^4 \log^2(|B_s|/\epsilon) \log(\bar{q}), \tag{44}$$

where we use $\Delta t \bar{q} = \mathcal{O}(1)$. The number of the unitary operators of $U_{j,B_s}$ is proportional to

$$\frac{|\Lambda|}{|B_s|} \cdot \frac{t}{\Delta t}. \tag{45}$$

Therefore, by combining the estimations of (44) and (45), the gate complexity implements $\{U_{j,B_s}\}_{j,s}$ is given by

$$\frac{t\ell^{3D}|\Lambda|}{\Delta t} \log^2(\ell^D/\epsilon) \log(\bar{q}) = |\Lambda|t\bar{q} \cdot \text{polylog}(|\Lambda|t\bar{q}/\epsilon),$$

where we use $\Delta t = \mathcal{O}(1/\bar{q})$ and $|B_s| = \mathcal{O}(\ell^D)$. We obtained the same estimation when implementing $\{U_{j,B_{s,s+1}}\}_{j,s}$. Therefore, we obtain the desired gate complexity to implement the unitary operator $U_{\Lambda}(0 \to t)$.

## Data availability
Data sharing does not apply to this paper, as no datasets were generated or analyzed during the current study.

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

## Acknowledgements

T.K. acknowledges Hakubi projects of RIKEN and was supported by Japan Society for the Promotion of Science KAKENHI (Grant No. 18K13475) and Japan Science and Technology Agency Precursory Research for Embryonic Science and Technology (Grant No. JPMJPR2116). K. S. was supported by JSPS Grants-in-Aid for Scientific Research (No. JP19H05603, No. JP19H05791 and No. JP23H01099).

## Author contributions

T.K., T.V.V., and K.S. contributed to the conception of the work, the analysis and interpretation, and the preparation and revision of the manuscript.

## Competing interests

The authors declare no competing interests.
