## [Peer Review File · Nature Communications]

Effective light cone and digital quantum simulation of interacting bosonsREVIEWER COMMENTS

Reviewer #1 (Remarks to the Author):

----- Summary -----

The authors expand on a past series of works to derive Lieb-Robinson (LR) bounds for bosonic Hamiltonians of the Bose-Hubbard form. They prove that: i) transport of bosons is confined to within a nearly-linear light cone; ii) propagation of arbitrary information lies within a super-ballistic front scaling as t^D (with D being spatial dimension); iii) these results imply efficient quantum simulation of such systems. The first result is (if I've understood correctly) completely general, but the latter two assume a "low boson density" condition. Nonetheless, they still have a much wider scope than past works, since these results hold for otherwise arbitrary initial states. Furthermore, the authors demonstrate by explicit example that their t^D scaling of the optimal information front is tight.

I should say before proceeding: I have not checked over the quite lengthy proof itself, rather merely the main text and the associated "Method" appendix (which I greatly appreciated the inclusion of, by the way). That said, I can confirm that the structure of the proof makes perfect sense (although I do have some lingering questions), so I am willing to trust that the technical analysis is correct as well.

----- Evaluation -----

I'm torn on whether to recommend this paper for publication. On the one hand, it's an impressive technical accomplishment that yields some interesting conceptual results as well (such as the separation between particle transport and information propagation). On the other hand, I think that the caveats to the results significantly lessen its impact. I discuss all this below. Since I'm not sure to what extent to balance the two, I think I'll stop here — with separate assessments of the scientific content and its potential impact — and leave it to the editor to decide.

Like I said, this work is an impressive technical achievement. It has been notoriously difficult

to derive Lieb-Robinson bounds for bosonic systems, and with the authors' results, we now have a good understanding of optimal LR bounds in a wide variety of experimentally relevant situations. Furthermore, it is quite interesting that these bounds imply information propagation can behave very differently than particle transport.

That said, I don't think it's fair to claim that these results settle the problem once and for all, nor do they provide a radically new way of studying bosonic many-body systems:

- While Bose-Hubbard Hamiltonians are indeed particularly relevant in experiments, they are not the only ones — lots of scattering processes can change the internal states of the particles involved. Even if more general Hamiltonians can lead to extremely rapid propagation, it will be important to characterize their speed limits as well.

- The assumption of low particle number per site, while also reasonable, puts this work in a similar vein as past studies which already proved LR bounds for finite-density steady states. In other words, given those past works, I don't think anyone would be surprised by the existence of a more general LR bound under these conditions. (In this sense the fact that information propagation scales as t^D rather than t is the most surprising aspect, but here I'll admit I'm missing any sort of physical interpretation — it would strengthen the case for this paper a lot if the authors could provide a more physical way of understanding this result.)

- On a related note, to the extent I understand it (and again I have only gone carefully through the high-level overview), the proof here does rely heavily on ideas already established in past works. So I don't think it's providing a particularly novel perspective in that sense (of course the authors should correct me if I'm wrong).

- Lastly, the additional logarithmic factors in the LR light cones very well could be removed by a more refined analysis.

Again, all of this is not to detract from the technical accomplishments. It's just to say that I could easily imagine follow-up works that would be justified in claiming this one is not the end of the story after all in some meaningful ways.

----- Comments -----

Regardless of the editor's decision to accept or reject, I do have some additional comments & questions for the authors that I feel would help improve the presentation.

1) Any sense of the tightness of Result 1? Specifically in situations which don't obey the "low-boson density" condition (Eq 3)?

2) To better connect the protocol in Fig 4 with Result 2, I think it would be helpful to explicitly state what the operator O_X is that propagates as t^D in the protocol. Is it the phase operator $e^{i n_X}$ alluded to at a different point in the paper?

3) I'm probably just missing something simple, but I'm wondering why the protocol in Fig 4 can't be modified to cause particle transport as well over a distance t^D . Consider an "information path" that, instead of having two horizontal sites per vertical coordinate, has only a single site. Label the sites of the information path by y (the vertical coordinate). After causing t^{D-1} particles to pile up on each site y exactly as you do, can I not use your ideas to get particles moving down the information path at a velocity t^{D-1} :

- Moving sequentially down the path, use infinitely large Hubbard interactions to project out the two states $|n_y, n_{y+1}\rangle$ and $|n_y - 1, n_{y+1} + 1\rangle$.
- The hopping term that couples those two states has matrix element $J \sqrt{n_y n_{y+1}}$, meaning we can evolve $|n_y, n_{y+1}\rangle$ into $|n_y - 1, n_{y+1} + 1\rangle$ in a time proportional to $1 / \sqrt{n_y n_{y+1}} \sim t^{-D+1}$.
- Thus by working sequentially down the path, we can move one particle from site 0 to site $R \sim t^D$ in a time t .

Is it clear what I mean? It's analogous to what you do, just moving a particle vertically rather than horizontally. Like I said, I'm probably just missing something, but this has gotten me thinking about how to reconcile your construction with the much slower particle transport that you proved in Result 1. If you can shed some insight on this, it might be worth adding to the paper.

4) In your outline of the proof for Result 1, the factor $\delta_l \sim e^{-l}$ still multiplies $n_X[l]$ in Eq 13. So for $l = O(1)$, don't you still get an exponential prefactor $(1 + \delta_l)^t$? Is this actually where the logarithmic correction to the light cone comes from, i.e., you really take $l = O(\log\{t\})$?

5) I'm afraid I don't see how the text around Eq 21 enters into any of the subsequent discussion. Performing site-dependent boson number truncation makes a lot of sense, but why is it important that the projected Hamiltonian obeys Eq 21? Perhaps I just need to see what g_0 and g_1 look like for the problem at hand.

6) I'm confused by how to use the projectors P_s in deriving the optimal light cone for Result 2, i.e., the text around Eq 25 (although I understand Eq 25 itself). The set $P_s = |\psi_{i,q}\rangle\langle\psi_{i,q}|$ seems very tailored to your specific example. How do you choose P_s more generally?

7) There's a lot of free parameters in Eq 26, and I think it'd be good to state explicitly what they end up being. For example, should l be $O(1)$ or scale as t ? Does Q scale as R ? Even if you don't explain where these choices come from, it would help a lot when unpacking the argument and seeing how it ultimately leads to the result $R \sim t^D$.

Reviewer #2 (Remarks to the Author):

REFeree REPORT FOR NATURE COMMUNICATIONS

Optimal light cone and digital quantum simulation of interacting bosons

by T. Kuwahara, T.V. Vu, and K. Saito

Summary. The paper under review studies the propagation of particles and information in systems of interacting bosons hopping on a lattice, specifically, the Bose-Hubbard model. This model is of practical relevance because lattice boson Hamiltonians can be created in the laboratory through optical lattices which are highly tunable. An important question, both from a theoretical and practical viewpoint, is whether these models satisfy a *Lieb-Robinson bound*, which would say that information propagation is approximately restricted to a light cone. This mimicking of the strict causality of relativistic theories is well-known for quantum spin systems and lattice fermions. The standard arguments break down for lattice bosons which have unbounded local Hilbert space dimension and interaction strength. As the authors mention, this question is practically relevant e.g., for efficiency guarantees of boson simulation.

There has recently been significant interest and activity in the many-body community to address this question and several partial results have been obtained by various groups. However, all of the prior results either require strong, non-generic assumptions on the initial state or they only control propagation in some weaker sense. This paper is the first one to address this problem by providing useful bounds on transport and information propagation. The particle propagation bound holds for all initial states and the information propagation bound (a.k.a. Lieb-Robinson bound) holds under a physically very reasonable bound on the particle density. The authors achieve this by an analytical tour-de-force (contained in well over 100 pages of Supplemental Material) where they develop various novel ideas. For example, (i) to estimate the particle number moments, they obtain tight control on a hierarchy of differential inequalities and (ii) to derive the Lieb-Robinson bound, they introduce a refinement which takes into account average-case bounds on the interaction strength. The latter in particular has a good chance of becoming a useful tool in other contexts.

Overall, I consider this work an impressive breakthrough on a highly relevant and active research problem. The work contains both conceptual and technical innovations that push the area forward.

Issues to be addressed. My main issue with the current manuscript is that the authors assert many times that the t^D -shaped light cone they derive for information propagation (cf. Result 2) is “optimal”. First, this is obviously found in the title. Moreover, from the abstract:

This study reveals an optimal light cone to limit the information propagation in interacting bosons, where the shape of the effective light cone depends on the spatial dimension.(...) The results of this study settle the notoriously challenging problem...”

The first sentence of the Outlook section reads:

“Our study clarified the qualitatively optimal light cone for systems with Bose-Hubbard type Hamiltonian.”

However, the truth is that the example Hamiltonian that achieves such a t^D -shaped light cone is

non-generic and time-dependent in a rather artificial way. The Hamiltonians achieve the light cone because certain hopping terms are discontinuously turned off. Consequently, it is unclear to what extent the conclusions are reproducible for the actual Bose-Hubbard Hamiltonian (more below).

Two further comments on this:

- There is a possible interpretation of “optimal” as meaning “optimal within the wider class of time-dependent Bose-Hubbard Hamiltonians”. However, with the current abstract and first paragraphs, anyone who does not read the paper closely will be left with the impression that the bound is optimal for the standard, time-independent Bose-Hubbard Hamiltonian and that’s not what’s shown here. Moreover, the example Hamiltonian is discontinuous in time, which likely does not fall into the range of validity of the bounds (which is unspecified in the current manuscript).
- The authors expect that the mechanism they identify for the special choice of time-dependent Hamiltonian will also lead to the same information propagation in the standard and physically relevant Bose-Hubbard Hamiltonian. However, this is not at all clear at this stage. First, one needs to make sure that the state with high-boson density \bar{n} on the “1D information path” indeed diffuses the bosons to an $O(1)$ density in time $\sim \bar{n}^{\frac{1}{D-1}}$. If this is true, then one can create initial states that form these 1D information paths. Then, one needs to check that before these bosons diffuse again, they can create sustained rapid information transport for reasonably large times. I believe that the best one can hope to prove from the construction given here is that there will be special initial states and intermediate state-dependent time scales on which information propagation can fill the t^D -shaped light cone. (The paper itself does not discuss any specific route to proving optimality for the usual Bose-Hubbard Hamiltonian.)

The limitations of the optimality result described above are hard to find in the current presentation, especially towards the beginning of the paper. Readers will easily get the impression that the question of optimality is settled for the standard Bose-Hubbard Hamiltonian when in fact it is not. The authors present a reasonable mechanism for fast information propagation in an artificial toy model, but they do not show that it has similar implications for the standard Bose-Hubbard Hamiltonian. This paper has plenty of other interesting insights also without discussing optimality the way it is currently done.

Therefore, I suggest to reword the title, the abstract, and the introduction to make it clear that the information propagation bound is only shown to be achieved for a special choice of time-dependent Hamiltonian.

Recommendation. If the authors revise the presentation (including title, abstract, and outlook section) in a way that makes the constraints of the derived optimality result significantly clearer, then I strongly recommend the paper for publication in *Nature Communications*.

Additional comments.

Below is a list of minor corrections and comments for the consideration of the authors.

1. When the authors revise the presentation of the optimality result, they should also compare the mechanism they identify for the time-dependent toy Hamiltonian with the mechanism found by Eisert-Gross for supersonic information propagation even in 1D.

2. On p. 2 the sentence “The function $f(\{\hat{n}_i\}_{i \in \Lambda})$ includes arbitrary long-range boson-boson couplings.” can be easily misunderstood and should be changed to “The function $f(\{\hat{n}_i\}_{i \in \Lambda})$ includes **arbitrarily long-ranged** boson-boson **density** couplings.” The same applies to a similar sentence shortly after Result 1.
3. On p. 2 when writing “Moreover, all results are applied to the time-dependent Hamiltonians.” the authors should briefly specify what kind of time dependence is allowed. (E.g., the coefficients are taken to be smooth functions in t .)
4. On p. 3, the authors write “As discussed in Result 2, the speed of information propagation is proportional to t^{D-1} .” This implicitly assumes optimality which is not proved in Result 2. The sentence should be changed to “As discussed in Result 2, the **bound on the** speed of information propagation is proportional to t^{D-1} .”
5. After eq. (5) a space is missing after “circuit”.
6. In the paragraph after (11), τ and t get mixed up a few times. I believe t should actually be replaced by τ in a few places.
7. On p. 6, the estimate $\text{tr}(\rho \hat{\mathcal{D}}_{X[\ell]}) \approx e^{-\mathcal{O}(\ell)} \text{tr}(\rho \hat{n}_X)$ does not seem quite correct. First, $e^{-\mathcal{O}(\ell)}$ is not good notation because $\mathcal{O}(\ell)$ just means growing *at most* like ℓ , better would be $\mathcal{O}(e^{-C\ell})$. (This confusion also appears in several places in the SM. This is normally resolved by writing Ω for lower bounds and \mathcal{O} for upper bounds.) Second, it is not clear why this estimate should hold. This would be the case if the j -sum in the definition of $\hat{\mathcal{D}}_X$ was restricted to X but it is not, it goes over Λ . So what happened to the particles near the surface of $X[\ell]$? In the Appendix, it seems like one only gets decay after solving the optimization problem over $0 \leq \ell \leq r$. Finally, there’s a hat over \mathcal{D} missing in the next line.

Reviewer #3 (Remarks to the Author):

Summary

In the manuscript NCOMMS-23-00767-T, the authors derive bounds on the spread of information in bosonic lattice systems. These bounds are notoriously hard to derive and are in some cases impossible. The authors get around the known limitations by assuming the well-behaved nature of the initial boson number distribution. Unlike other works, they do not need to make an assumption that the initial state is steady. The "well-behavedness" of the boson number distribution is bootstrapped by showing that if it is valid at time t , it is also valid at time $t+\Delta t$ (albeit to a lesser degree).

The authors show 3 main results. The first concerns the spread of bosons in a lattice system when there are arbitrary number-conserving interactions. The authors show that this speed of boson propagation grows at most logarithmically in time.

The second result concerns the spread of information rather than the spread of bosons. In this setting, the authors make an additional assumption on the initial boson number distribution (namely that the probability of observing a certain number of bosons at each site decays at least subexponentially with boson number). In this context, they observe that the speed of information propagation is now dimension-dependent. Specifically, they find that the speed is proportional to t^{D-1} in D dimensions (up to logarithmic corrections). This is a surprising result, since fermionic and spin systems have no such dimension-dependence of the speed (at least for short-range interacting systems). Nevertheless, the authors show that this behaviour is actually tight, instead of merely being an artifact of a possibly correct but loose bound. Specifically, they show that there is a state transfer protocol that achieves this speed of information propagation (although, see below for my questions about the proof).

Lastly, the authors derive an analogue of the result by Haah-Hastings-Kothari-Low on the gate count of simulating the bosonic system in time. For a very general form of the bosonic interactions, they obtain an upper bound for simulating a system for time t . This upper bound scales as t^{D+1} , which the authors surmise is not optimal and can be improved to t^D .

Evaluation

The paper is extremely well-written and gives a high-level overview of the techniques, the subtleties involved, and walks the reader through an easier version of the proof in the Methods section before relegating the more powerful version using the same techniques to the Supplementary Information. It also covers very well the ideas used by other papers preceding it and gives a very good feel for the techniques.

I do have one question about the tightness result of Result 2, which I hope the authors can explain/respond to my satisfaction in order to convince me of its correctness. Barring this potential issue, I believe the rest of the manuscript is correct to the extent that I have read it. The issue is the following.

In the section where they argue for the optimality of the light cone, the authors claim that they realise a CNOT operation in a time that scales as $1/(\bar{J}\bar{n}_t)$ or as $1/(\bar{J}^D t^{D-1})$. I found the justification for this in the Methods section incomplete. According to the authors' explanation, the operations in Eq. (27) are performed in two stages, one where they apply the free boson evolution to achieve $|N,1\rangle \rightarrow |1,N\rangle$, and another where they transform the state $|1,N\rangle \rightarrow |0,N+1\rangle$. In the first step, the time taken to achieve the transformation should be $\sim 1/(\bar{J})$ since one is transforming the entire creation operator at site 1 to a creation operator at site 2 (and is thus transferring all the bosons on site 1 to site 2). And because this takes time $O(1)$, even if there is a strong bosonic enhancement in the second step, the overall time taken is still $O(1)$. So I don't understand how the authors claim that the time taken for a CNOT scales inversely with the boson number as mentioned. I hope there is a reasonable explanation here that surmounts this difficulty.

Minor Comments

- While the figures are very illuminating, Fig. 1 is not referred to in the main text.
- On page 1, the authors refer to two conditions for getting Lieb-Robinson bounds as (i) and (ii). On page 2, they refer to two primary targets i) and ii) separately. In the last line of the first paragraph of page 2 beginning "Until now, we are far from the long-sought goal of characterizing the optimal forms of the effective light cones for the speed of i) and ii)", it is

unclear that they are referring to the primary targets and not the two conditions introduced on page 1. Please use other letters such as (a), (b) to avoid confusion.

- Just after Eq. (1), the authors say "The function $f(\{\hat{n}_i\}_{i \in \Lambda})$ includes arbitrary long-range boson-boson couplings". Please say whether f is also arbitrarily long-range and whether there is any bound on the strength of f . Please also give some intuition why the arbitrary long-ranged nature of f does not matter. I suspect it is because it commutes with the boson number but a clarification would be nice. Is a non k -local interaction, for example of the form $\prod_i n_i$ (the product of all boson numbers) also allowed?

- In Result 1, please again specify whether the constants depend on f or some notion of the strength of f in some way.

- There are typos in Kothari's name (multiple instances).

- Page 5, "The gate complexity of the quantum simulation has room for qualitative improvement from $|\Lambda|t^{D+1}\text{polylog}(|\Lambda|t/\epsilon)$ to $|\Lambda|t^D\text{polylog}(|\Lambda|t/\epsilon)$: what is this expectation based on? I don't know if there are any lower bounds known, are there?

- Page 6, in the definition of D_X , is the sum over j really over the whole lattice Λ ?

- In the equation after Eq. (11) (please consider numbering all the equations, since otherwise it is hard to refer to them), I believe the first argument should be τ instead of t .

- After Eq. (12), when the authors say "By solving the above optimization problem appropriately", it would be nice to say what the optimal ℓ is. Or at least a mention of how it scales with problem parameters.

- Page 7, "...the time evolution of the boson number operator \hat{n}_i is roughly upper-bounded by the boson number on the ball region $i[l_t]$, that is, $\hat{n}_i[l_t]$, where $l_t = O(t \log t)$ ": what about the third term in Result 1 that goes as $(\dots + ts)^\alpha$? Is it ignored here? But that is presumably not small always.

- Before Eq.(23), should it be $\hat{n}_j|\psi_{\{i,q\}}\rangle=0$ instead of j in the subscript?

- I did not understand the point made in this paragraph and how the sentence "Therefore, the superposition of different boson number configurations prohibits the site-dependent boson number truncation" followed from the paragraph. It makes sense that taking $q_i \geq q$ is in some sense necessary. How does that prohibit a site-dependent boson number truncation?

- Second column of page 8: I got the matrix elements $\langle 1,N|V|1,N\rangle = +U N(N+1)$ (instead of a minus sign in front) and the same for the other matrix element.

List of Changes

1. We added several explanations which reflect the comments from the reviewers. All the revised parts are highlighted by dark-red text.
2. We changed the title in accordance with the advice of the second reviewer.
3. We added one section in the supplementary information on the information propagation for time-independent Hamiltonians.
4. We added one section in the Method section to resolve the ambiguity of the particle transport in response to the first reviewer.

Reply to Reviewer #1

Summary

The authors expand on a past series of works to derive Lieb-Robinson (LR) bounds for bosonic Hamiltonians of the Bose-Hubbard form. They prove that: i) transport of bosons is confined to within a nearly-linear light cone; ii) propagation of arbitrary information lies within a super-ballistic front scaling as t^D (with D being spatial dimension); iii) these results imply efficient quantum simulation of such systems. The first result is (if I've understood correctly) completely general, but the latter two assume a “low boson density” condition. Nonetheless, they still have a much wider scope than past works, since these results hold for otherwise arbitrary initial states. Furthermore, the authors demonstrate by explicit example that their t^D scaling of the optimal information front is tight.

I should say before proceeding: I have not checked over the quite lengthy proof itself, rather merely the main text and the associated “Method” appendix (which I greatly appreciated the inclusion of, by the way). That said, I can confirm that the structure of the proof makes perfect sense (although I do have some lingering questions), so I am willing to trust that the technical analysis is correct as well.

Reply: We would like to thank you for your careful reading and the very insightful comments on this work. All the comments are quite helpful for us to improve the main part of the paper, in particular, regarding the discussion on the particle transport problem. Following your suggestions, we have substantially rewritten the manuscript in the main part. In the following, please let us address the additional comments one by one. The revised parts are highlighted by the dark-red texts in the new manuscript.

Evaluation

I'm torn on whether to recommend this paper for publication. On the one hand, it's an impressive technical accomplishment that yields some interesting conceptual results as well (such as the separation between particle transport and information propagation). On the other hand, I think that the caveats to the results significantly lessen its impact. I discuss all this below. Since I'm not sure to what extent to balance the two, I think I'll stop here — with separate assessments of the scientific content and its potential impact — and leave it to the editor to decide.

Like I said, this work is an impressive technical achievement. It has been notoriously difficult to derive Lieb-Robinson bounds for bosonic systems, and with the authors' results, we now have a good understanding of optimal LR bounds in a wide variety of experimentally relevant situations. Furthermore, it is quite interesting that these bounds imply information propagation can behave very differently than particle transport.

That said, I don't think it's fair to claim that these results settle the problem once and for all, nor do they provide a radically new way of studying bosonic many-body systems:

Reply: Thank you for the positive comment! On the criticisms you mentioned, we would like to answer them one by one. We expect that they will reduce your concern about our work.

- While Bose-Hubbard Hamiltonians are indeed particularly relevant in experiments, they are not the only ones — lots of scattering processes can change the internal states of the particles involved. Even if more general Hamiltonians can lead to extremely rapid propagation, it will be important to characterize their speed limits as well.

Reply: Thank you for the comment! As you say, we can consider more general classes of interacting boson systems, and it poses an important future problem.

We have also considered the possibility of extending our current result to more general Hamiltonians. At this stage, we have reached the conclusion that generalized models have no speed limit unless further strong assumptions are imposed; here, ‘no speed limit’ means that information can reach to arbitrary distant positions by a finite time of $\mathcal{O}(1)$. For example, the simplest generalization to consider an interaction-induced tunneling terms [T. Sowiński, et al., *Phys. Rev. Lett.* **108**, 115301 (2012)], which converts the Bose-Hubbard model to

$$H = \sum_{\langle i,j \rangle} J_{ij} b_i^\dagger b_j + \sum_{\langle i,j \rangle} T_{ij} b_i^\dagger (\hat{n}_i + \hat{n}_j) b_j + \frac{1}{2} \sum_i U_i \hat{n}_i (\hat{n}_i - 1) + \sum_i \mu_i \hat{n}_i. \quad (\text{R.1})$$

Unfortunately, at this level of generalization, we can no longer prove a speed limit in general even if we start from the Mott state, which has been shown in Supplementary material in our subsequent paper [Tan Van Vu, Tomotaka Kuwahara, and Keiji Saito, [arXiv:2307.01059](https://arxiv.org/abs/2307.01059)].

This point forces us to reconsider natural additional conditions to the system or initial states beyond the low-boson-density condition, which has been used in the study of bosonic Lieb-Robinson bound. The problem is beyond the scope of this study, but we would like to add one paragraph to mention this point in the outlook section as follows:

“As a straightforward extension, it is intriguing to investigate under what conditions boson-boson interactions like $b_{i_1} b_{i_2} b_{i_3}^\dagger b_{i_4}^\dagger$ can lead to information propagation with a limited speed. In this scenario, relying solely on the low boson density condition proves insufficient for regulating the speed of information propagation (cf. Supplemental Material in Ref. [52]). ”

- The assumption of low particle number per site, while also reasonable, puts this work in a similar vein as past studies which already proved LR bounds for finite-density steady states. In other words, given those past works, I don’t think anyone would be surprised by the existence of a more general LR bound under these conditions. (In this sense the fact that information propagation scales as t^D rather than t is the most surprising aspect, but here I’ll admit I’m missing any sort of physical interpretation — it would strengthen the case for this paper a lot if the authors could provide a more physical way of understanding this result.)

Reply: Thank you for the comment! This point may be relevant to the previous reply. Without

any assumption on the boson number, we have no speed limit for information propagation in general [P. Barmettler, et al., *Phys. Rev. A* 85, 053625 (2012)]; that is, the speed of information propagation is proportional to the number of bosons at local sites. This is the main reason that the low-boson-density condition is the minimal assumption to derive a meaningful Lieb-Robinson bound. It is also worth noting that the speed of the particle transport is NOT influenced by the boson numbers at each site, which makes another clear difference between the particle transport and the information propagation. We change several sentences and add the explanation on this point:

We emphasize that without assuming any conditions for the boson number, there is no speed limit for general information propagation. More precisely, the speed of information propagation is directly proportional to the number of bosons at local sites [41]. This underscores why the low-boson-density condition is the minimal assumption required to establish a meaningful Lieb-Robinson bound. This point also makes a clear difference between the information propagation and the particle transport, in which no conditions are imposed for initial states in Result 1.

As you mentioned, the non-linearity of the light cone is the most non-trivial point that makes a difference from the previous works, which imposes the steady-state-condition as an additional strong assumption [T. Kuwahara and K. Saito, *Phys. Rev. Lett.* 127, 070403 (2021)] and [C. Yin and A. Lucas, *Phys. Rev. X*, 12, 021039 (2022)]. We believe that our result shed new light on the study of information propagation in the sense that it poses a non-intuitive phenomenon^{*1}.

Furthermore, the physical interpretation of the accelerating information is understood as follows. The first point is that the velocity of information propagation is proportional to the number of bosons at local sites, as has been shown in [P. Barmettler, et al., *Phys. Rev. A* 85, 053625 (2012)]. Therefore, if dynamics enhances boson concentrations in specific one-dimensional regions, the density of bosons in this 1D region increases with time. This in turn leads to the acceleration of the information propagation on this 1D ‘information path’. It only happens in high-dimensional systems, because in one-dimensional systems, if bosons concentrate onto a specific region, the surrounding region has sparse boson densities, which prohibits the persistent acceleration. We added one paragraph just before the section on “Optimality of the effective light cone.”

The accelerated information propagation can be interpreted physically as follows. According to Ref. [41], the velocity of information propagation is directly proportional to the number of bosons at local sites in the presence of boson-boson interactions. Consequently, if dynamic processes cause an increase in boson concentrations within specific one-dimensional regions, the boson density in that 1D region will rise over time. As a result, information propagation on this 1D “information path” experiences acceleration. This phenomenon is specific to high-dimensional systems. In one-dimensional systems, if bosons concentrate in a specific region, the surrounding areas exhibit sparse

^{*1} In page 26 of the paper [C. Yin and A. Lucas, *Phys. Rev. X*, 12, 021039 (2022)], the authors say “We expect that no state with superluminal propagation exists; however, techniques that combine ours with those of Ref. [30] may be required to definitively resolve this issue.”.

boson densities, preventing persistent acceleration.

- On a related note, to the extent I understand it (and again I have only gone carefully through the high-level overview), the proof here does rely heavily on ideas already established in past works. So I don't think it's providing a particularly novel perspective in that sense (of course the authors should correct me if I'm wrong).

Reply: Thank you for the comment! It is always difficult to argue what is technically *novel*. Without the novelty of the used techniques, we believe that the understanding of new physics is more important than the technical contribution.

But, the following points have not been considered anywhere before.

1. To derive the Result 1, we have to overcome the long-standing caveat in the result by Schuch, Harrison, Osborne, and Eisert [N. Schuch, et al., *Phys. Rev. A* 84, 032309 (2011)], which provided the first general result on the information propagation in the Bose-Hubbard model. In our analyses, one of the key techniques is to adopt parameters as degrees of freedom in the upper bound and consider the optimization problem for all the parameters (e.g., Sec. S III E in the supplemental), where the parameters are defined by the length scales to characterize the subsets. Parametrizing the upper bound and solving the optimization problem may be a useful technique in the future works.
2. After obtaining Result 1, a simple use of the previous techniques ^{*2} yields the light cone shape of $R \propto t^{D+1}$ instead of $R \propto t^D$, which has been emphasized in Sec. S VII in the supplementary materials.

For the refinement of the optimal light cone, we resolved the following challenging problem. We have to take in the point that the boson concentration cannot occur simultaneously. That is, even if the bosons clump together, not all the sites have large boson numbers, and hence the boson number is on average much smaller than the boson number in the worst case. But, the primary difficulty is that the configuration of the boson densities is not fixed. A quantum state includes various configurations as superposition, as has been depicted in Fig. 6 in the main manuscript. This prohibits the simple boson number truncations in proving the optimal Lieb-Robinson bound. As far as we know, there are no mathematical tools to treat such situations. To resolve the problem, various new ideas and a substantial amount of lemmas (from Sec. S VIII to Sec. S X, over 40 pages) were necessitated.

From the above points, we would like to appeal that our result is not merely a straightforward connection of existing techniques, but we need to develop various new analytical tools to achieve

^{*2} We mean the effective Hamiltonian theory that is based on the boson number truncation, which plays a crucial role in our previous work [T. Kuwahara and K. Saito, *Phys. Rev. Lett.* 127, 070403 (2021)].

a significant jump from the previous analytical limit.

- Lastly, the additional logarithmic factors in the LR light cones very well could be removed by a more refined analysis. Again, all of this is not to detract from the technical accomplishments. It's just to say that I could easily imagine follow-up works that would be justified in claiming this one is not the end of the story after all in some meaningful ways.

Reply: Thank you for your comment! While it would be preferable to eliminate logarithmic factors, doing so would significantly complicate the current analyses. Up to this point, we have treated the logarithmic correction as a non-essential factor. However, it is important to acknowledge that our current analyses have limitations in removing logarithmic factors. Actually, we have included this point to the future problem in the outlook of the previous version, and we rewrite the sentences as follows:

First, the obtained bounds incorporate logarithmic corrections, but there is a possibility of their removal through a refinement of the current analyses. Currently, it does not seem to be a straightforward problem to remove them using our existing techniques. A possible starting point to achieve this is to consider the difference between average values as $\text{tr} \left[\left(O_{X_0}(t) - O_{X_0}(H_{X_0[R]}, t) \right) \rho_0 \right]$ instead of the trace norm $\left\| \left(O_{X_0}(t) - O_{X_0}(H_{X_0[R]}, t) \right) \rho_0 \right\|_1$. While this quantity may not capture the propagation of *total information*, a significantly stronger bound can be proven in one-dimensional systems, where the light cone form strictly follows a linear form with time [49]. Through the refinement and combination of existing techniques, there is a possibility of eliminating the logarithmic corrections in our current bound in future studies.

2) To better connect the protocol in Fig 4 with Result 2, I think it would be helpful to explicitly state what the operator O_X is that propagates as t^D in the protocol. Is it the phase operator e^{inx} alluded to at a different point in the paper?

Reply: Thank you for your comment! We considered to provide an explicit form, but it seems that it is not possible to give a simple example. Instead, we explain that a sequence of CNOT operations necessarily implies the operator spreading from the discussion in [Bravyi, Hastings, and Verstraete, Phys. Rev. Lett. 97, 050401 (2006)].

Let us consider an initial product state $|0\rangle^{\otimes n}$. Then, by flipping the endmost qubit by $|1\rangle$, then the $(n - 1)$ -sequential CNOT operations transform the state to $|1\rangle^{\otimes n}$, whereas without the flipping, the state is unchanging. Hence, by encoding the information by flipping or non-flipping the endmost qubit, one can send the information through the sequence of CNOT operations. From [Bravyi, Hastings, and Verstraete, Phys. Rev. Lett. 97, 050401 (2006)], this process necessarily induces the operator spreading of the flipping unitary to the endmost qubit in the Heisenberg picture.

We add the following paragraph in the main text:

One can demonstrate that the number of CNOT operations is directly linked to the distance of

the operator spread, following the discussion in Ref. [25]. To illustrate this, consider two types of operations: flipping or non-flipping the endmost qubit on the information path at the time $t/2$, where the boson concentration has been over. By flipping the endmost qubit to $|1\rangle$, m -sequential CNOT operations transform the state to $|1\rangle^{\otimes m}|0\rangle^{\otimes \ell-m}$, while without flipping, the state remains unchanged $|0\rangle^{\otimes \ell}$. Here, ℓ represents the total length of the information path. By encoding classical information as flipping ($= 0$) or non-flipping ($= 1$) of the endmost qubit, one can transmit 1 bit of information through the sequence of CNOT operations. The connection between Holevo capacity and operator spreading [25] implies that this process necessarily induces the operator spreading of the flipping unitary to the endmost qubit in the Heisenberg picture. Thus, $\bar{J}^D t^D$ CNOT operations during the time t achieves the Lieb-Robinson velocity of $\bar{J}^D t^{D-1}$. This accelerating information propagation must be clearly distinguished from boson transport with a constant velocity, as in Result 1.

3) I'm probably just missing something simple, but I'm wondering why the protocol in Fig 4 can't be modified to cause particle transport as well over a distance t^D . Consider an "information path" that, instead of having two horizontal sites per vertical coordinate, has only a single site. Label the sites of the information path by y (the vertical coordinate). After causing t^{D-1} particles to pile up on each site y exactly as you do, can I not use your ideas to get particles moving down the information path at a velocity t^{D-1} : - Moving sequentially down the path, use infinitely large Hubbard interactions to project out the two states $|n_y, n_{y+1}\rangle$ and $|n_y - 1, n_{y+1} + 1\rangle$. - The hopping term that couples those two states has matrix element $J\sqrt{n_y n_{y+1}}$, meaning we can evolve $|n_y, n_{y+1}\rangle$ into $|n_y - 1, n_{y+1} + 1\rangle$ in a time proportional to $1/\sqrt{n_y n_{y+1}} \sim t^{-D+1}$. - Thus by working sequentially down the path, we can move one particle from site 0 to site $R \sim t^D$ in a time t . Is it clear what I mean? It's analogous to what you do, just moving a particle vertically rather than horizontally. Like I said, I'm probably just missing something, but this has gotten me thinking about how to reconcile your construction with the much slower particle transport that you proved in Result 1. If you can shed some insight on this, it might be worth adding to the paper.

Reply: Thank you for the insightful comments. Using our protocol, we can achieve the transformation of

$$(b_i^\dagger)^m |\text{Mott}\rangle \rightarrow (b_j^\dagger)^m |\text{Mott}\rangle \quad (\text{R.2})$$

by a time evolution as long as $d_{i,j} \lesssim t^D$, where $|\text{Mott}\rangle$ is the Mott state with one boson at each of the sites. While this process may seem to imply particle transport over a distance $d_{i,j}$, it is crucial to recognize that bosons are indistinguishable particles. Thus, the above process does NOT imply genuine particle transport. For instance, without using our protocol, the process (R.2) can be achieved in a constant time for arbitrary distances (Fig. 2). To characterize particle transport, we must ensure that the increased bosons indeed come from the distant region. This assurance be achieved in the following cases:

1. If $\text{tr}(\rho(t)\hat{n}_X) > \text{tr}(\rho\hat{n}_{X[R]})$, we can ensure that a part of the increase in boson number comes from the region $X[R]^c$, achieving particle transport over a distance R .

FIG. 1. Transformation from $b_1^\dagger|\text{Mott}\rangle$ to $b_n^\dagger|\text{Mott}\rangle$ (a). This process is equivalent to one hopping from left to right of all bosons (b), where the left-end two sites and the right-end two sites are merged into one site, respectively. Then, this process takes time of $\mathcal{O}(1)$ for arbitrarily long 1D chains.

2. By making particular bosons distinguishable from others (e.g., bosons with the spin degree of freedom), particle transport can be clearly defined.

The first case is addressed in Result 1, where we establish a finite speed. In the second case, we also prove the finite speed of transport by slightly generalizing Result 1. In this case, the Hamiltonian should also be generalized to

$$H = \sum_{\sigma} \sum_{\langle i,j \rangle} J_{i,j} (b_{i,\sigma} b_{j,\sigma}^\dagger + \text{h.c.}) + f(\{\hat{n}_{i,\sigma}\}_{i \in \Lambda, \sigma}). \quad (\text{R.3})$$

Then, the same operator inequality as in Result 1 holds for $\hat{n}_{X,\sigma}(t)$ for corresponding spin degrees σ .

In the context of this discussion, a more phenomenological explanation to ensure the finite velocity of boson transport is through the particle current. The particle current operator $\hat{J}_{i,i+1}$ between the sites i and $i+1$ is defined as

$$\hat{J}_{i,i+1} := J(ib_i b_{i+1}^\dagger + \text{h.c.}), \quad (\text{R.4})$$

where we consider a one-dimensional system for simplicity, and the free Hamiltonian is $H_0 = \sum_i J(b_i b_{i+1}^\dagger + \text{h.c.})$. Usually, the current is defined as the product of particle density and velocity, giving the speed of particle velocity as

$$v_{\text{transport}} \sim \left\| \frac{\hat{J}_{i,i+1}}{\hat{n}_i + \hat{n}_{i+1}} \right\| \leq 2J, \quad (\text{R.5})$$

where we use the operator inequality of $|\hat{J}_{i,i+1}| \preceq 2J(\hat{n}_i + \hat{n}_{i+1})$ from $|b_i b_{i+1}^\dagger| \leq \hat{n}_i + \hat{n}_{i+1}$ [see (S.459) in Supplementary Information]. Although it is non-trivial to derive our Result 1 only from this discussion, it provides a simple picture of why the speed of particle transport has a finite speed.

We added the above discussions in the Method section as “*Non-acceleration of Boson Transport*”.

4) In your outline of the proof for Result 1, the factor $\delta_\ell \sim e^{-\ell}$ still multiplies $n_X[\ell]$ in Eq 13. So for $\ell = O(1)$, don't you still get an exponential prefactor $(1 + \delta_\ell)^t$? Is this actually where the logarithmic correction to the light cone comes from, i.e., you really take $\ell = O(\log(t))$?

Reply: Thank you for your comment! That's correct, we assumed the condition of $\ell \gtrsim \log(t)$ at (S.281) in Theorem 1 in Supplementary Information. And the primary reason is to make $(1 + \delta_\ell)^t \lesssim t\delta_\ell$. We add one phrase to mention it:

“More precisely, the iterative use of the inequality (13) yields an additional coefficient $(1 + c_{\tau,1}\delta_\ell)^{t/\tau}$ to the first term $\hat{n}_{X[(t/\tau)\ell]}$ in (10). We need the condition $R \geq c_0 t \log t$ (or $\ell \propto \log(t)$) to ensure $(1 + c_{\tau,1}\delta_\ell)^{t/\tau} \lesssim 1 + c_{\tau,1}t\delta_\ell$.”

5) I'm afraid I don't see how the text around Eq 21 enters into any of the subsequent discussion. Performing site-dependent boson number truncation makes a lot of sense, but why is it important that the projected Hamiltonian obeys Eq 21? Perhaps I just need to see what g_0 and g_1 look like for the problem at hand.

Reply: Thank you for your comment! In the standard Lieb-Robinson bound, the Lieb-Robinson bound velocity is proportional to the maximum local energy, i.e.,

$$v_{\text{LR}} \propto \max_{i \in \Lambda} \sum_{Z: Z \ni i} \|h_Z\|. \quad (\text{R.6})$$

However, this estimation is too weak in deriving the bosonic Lieb-Robinson bound since the local energy depends on the boson number at the local site and can be as large as $\mathcal{O}(t^D)^{*3}$. This is why we introduce the constraints of Eq. 21, which implies that the average of the norms for the local interaction norms approaches \bar{g}_1 . In our situation, roughly speaking, the parameter \bar{g}_0 corresponds to the maximum local energy on one site, and \bar{g}_1 is the average local energy on one site.

I added some explanation on why we introduce the constraint Eq. 21 as follows:

To address the aforementioned point, we need to consider cases where the interaction strengths in a Hamiltonian depend on the locations. In the standard Lieb-Robinson bound, the Lieb-Robinson velocity is proportional to the maximum local energy [9,10]. However, this estimation is insufficient when deriving the bosonic Lieb-Robinson bound, as the local energy depends on the boson number at the local site and can be as large as $\mathcal{O}(t^D)$; our current goal is to derive the Lieb-Robinson velocity as t^{D-1} .

For this purpose, we consider a general Hamiltonian in the form of $H = \sum_{Z \subset \Lambda} h_Z$ with the additional constraint:

$$\frac{1}{m} \sum_{j=1}^m \|h_{Z_j}\| \leq \frac{\bar{g}_0}{m} + \bar{g}_1, \quad (\text{R.7})$$

where $\{h_{Z_j}\}_{j=1}^m$ are arbitrary interaction terms acting on subsets $\{Z_j\}_{j=1}^m$, respectively. Roughly speaking, the parameter \bar{g}_0 corresponds to the maximum local energy on one site, and \bar{g}_1 is the average local energy on one site. Under the above condition, each interaction term h_Z has an upper bound of $\bar{g}_0 + \bar{g}_1$. Thus, the standard Lieb-Robinson bound gives a Lieb-Robinson velocity of $\mathcal{O}(\bar{g}_0 + \bar{g}_1)$, which can be unfavorable if \bar{g}_0 is large. Through refined analyses, we can prove that the

^{*3} We here aim to derive the Lieb-Robinson velocity as t^{D-1} .

improved Lieb-Robinson velocity depends on the distance as $\mathcal{O}(\bar{g}_0/R) + \mathcal{O}(\bar{g}_1)$, eventually becoming $\mathcal{O}(\bar{g}_1)$ for sufficiently large R [Lemma 42 in Supplementary Information].

6) I'm confused by how to use the projectors P_s in deriving the optimal light cone for Result 2, i.e., the text around Eq 25 (although I understand Eq 25 itself). The set $P_s = |\psi_{i,q}\rangle\langle\psi_{i,q}|$ seems very tailored to your specific example. How do you choose P_s more generally?

Reply: Thank you for your comment! We realized that the discussion along the toy model is a bit confusing^{*4}, i.e.,

“To see the point, we define $|\psi_{i,q}\rangle$ as an arbitrary quantum state such that $\Pi_{i,q}|\psi_{i,q}\rangle = |\psi_{i,q}\rangle$ and $\hat{n}_j|\psi_{j,q}\rangle = 0$ ($j \in L, j \neq i$), where $\Pi_{i,q}$ has been introduced in (16). Then, for the superposition of

$$|\psi\rangle = |L|^{-1/2} \sum_{i \in L} |\psi_{i,q}\rangle, \quad (\text{R.8})$$

each site comprises q bosons with an amplitude of $|L|^{-1/2}$, and hence, the boson number truncation by $\bar{\Pi}_{L,\vec{q}}$ is justified only when $q_i \geq q$ for $\forall i \in L$. Therefore, the superposition of different boson number configurations prohibits the site-dependent boson number truncation. ”

Hence, we completely rewrite this part to make the point clearer as follows^{*5}:

“The primary challenge arises from the inability to obtain an accurate approximation for dynamics using the effective Hamiltonian $\bar{\Pi}_{L,\vec{q}}H\bar{\Pi}_{L,\vec{q}}$ for a specific choice of \vec{q} . This challenge is rooted in the superposition of quantum states with diverse boson configurations. For instance, consider a quantum state $|\psi\rangle$ represented as the superposition of two states, $|\psi_1\rangle$ and $|\psi_2\rangle$, where $\bar{\Pi}_{L,\vec{q}_1}|\psi_1\rangle = |\psi_1\rangle$ and $\bar{\Pi}_{L,\vec{q}_2}|\psi_2\rangle = |\psi_2\rangle$. Then, time evolution with the effective Hamiltonian $\bar{\Pi}_{L,\vec{q}_1}H\bar{\Pi}_{L,\vec{q}_1}$ provides a reliable approximation for $e^{-iHt}|\psi_1\rangle$ but not for $e^{-iHt}|\psi_2\rangle$. Conversely, time evolution with $\bar{\Pi}_{L,\vec{q}_2}H\bar{\Pi}_{L,\vec{q}_2}$ gives a good approximation for $e^{-iHt}|\psi_2\rangle$ but not for $e^{-iHt}|\psi_1\rangle$. Thus, a specific boson number truncation using $\bar{\Pi}_{L,\vec{q}}$ cannot be applied uniformly to all superposed states. It is necessary to consider different boson number truncations depending on the boson configuration of the superposed states, as discussed in Section S VIII A of the Supplementary Information. ”

Also, on the actual choice of P_s , the upper bound

$$\sum_{s=1}^M \left\| \left[O_X(\tau) - O_X(H_{X^{[\ell]}, \tau}) \right] \mathcal{P}_s \right\| \cdot \|\mathcal{P}_s \rho_0(t_1)\|_1. \quad (\text{R.9})$$

grows with the number of projections M . We cannot simply choose as the projection onto the Fock bases such as $\prod_{i \in L} \Pi_{i,q_i}$, which makes $M = e^{\mathcal{O}(L)}$. Hence, we need to choose a minimal set of $\{\mathcal{P}_s\}_{s=1}^M$ to meet our purpose. We have explicitly shown the choices in (S.572) and (S.573) in Supplementary Information.

^{*4} This point has been also pointed out by another reviewer.

^{*5} Please also see the last reply to the third reviewer.

In the revised manuscript, we added the following sentences:

Because the summation (R.9) increases with the number of projections M , we need to select a minimal set of $\{\mathcal{P}_s\}_{s=1}^M$ to achieve our goal. The choice of the projections is rather technical (see Sec. S.VIII A in Supplementary information).

7) There are a lot of free parameters in Eq 26, and I think it'd be good to state explicitly what they end up being. For example, should l be $O(1)$ or scale as t ? Does Q scale as R ? Even if you don't explain where these choices come from, it would help a lot when unpacking the argument and seeing how it ultimately leads to the result $R \sim t^D$.

Reply: Thank you for your comment! We have utilized 8 fundamental parameters in total, which were given in Table I in Supplementary Information. We mention this point at the beginning of the method section:

Throughout the proof, we denote $\mathcal{O}(1)$ as an arbitrary finite combination of the fundamental parameters, which are detailed in Table I of the Supplementary Information.

Also on the choice of Q , from (S.769), (S.773), and (S.774) we eventually choose it as

$$Q \propto \frac{t(R/t^D)^{\frac{D+1}{D}}}{\log^{\frac{D}{D-1}}(R/t)}. \quad (\text{R.10})$$

It would be complicated to show all the explicit choices of the parameters, and hence, we only refer to the equations in Supplementary Information that give the choices:

By choosing Q and τ appropriately and connecting the short-time evolution, we can prove the main statement (4) [see (S.769), (S.773), and (S.774) of Sec. S.IX.B in Supplementary Information].

Reply to Reviewer #2

Summary. The paper under review studies the propagation of particles and information in systems of interacting bosons hopping on a lattice, specifically, the Bose-Hubbard model. This model is of practical relevance because lattice boson Hamiltonians can be created in the laboratory through optical lattices which are highly tunable. An important question, both from a theoretical and practical viewpoint, is whether these models satisfy a Lieb-Robinson bound, which would say that information propagation is approximately restricted to a light cone. This mimicking of the strict causality of relativistic theories is well-known for quantum spin systems and lattice fermions. The standard arguments break down for lattice bosons which have unbounded local Hilbert space dimension and interaction strength. As the authors mention, this question is practically relevant e.g., for efficiency guarantees of boson simulation.

There has recently been significant interest and activity in the many-body community to address this question and several partial results have been obtained by various groups. However, all of the prior results either require strong, non-generic assumptions on the initial state or they only control propagation in some weaker sense. This paper is the first one to address this problem by providing useful bounds on transport and information propagation. The particle propagation bound holds for all initial states and the information propagation bound (a.k.a. Lieb-Robinson bound) holds under a physically very reasonable bound on the particle density. The authors achieve this by an analytical tour-de-force (contained in well over 100 pages of Supplemental Material) where they develop various novel ideas. For example, (i) to estimate the particle number moments, they obtain tight control in a hierarchy of differential inequalities and (ii) to derive the Lieb-Robinson bound, they introduce a refinement which takes into account average-case bounds on the interaction strength. The latter in particular has a good chance of becoming a useful tool in other contexts.

Overall, I consider this work an impressive breakthrough on a highly relevant and active research problem. The work contains both conceptual and technical innovations that push the area forward.

Reply: We would like to thank you for the positive assessment of the potential significance of our work. We have revised the manuscript so that the ambiguity is removed and the readability

is improved. The revised parts are highlighted by the dark-red texts in the new manuscript.

Issues to be addressed. My main issue with the current manuscript is that the authors assert many times that the t^D -shaped light cone they derive for information propagation (cf. Result 2) is “optimal”. First, this is obviously found in the title. Moreover, from the abstract:

This study reveals an optimal light cone to limit the information propagation in interacting bosons, where the shape of the effective light cone depends on the spatial dimension. (...) The results of this study settle the notoriously challenging problem...

The first sentence of the Outlook section reads:

“Our study clarified the qualitatively optimal light cone for systems with Bose-Hubbard type Hamiltonian.”

However, the truth is that the example Hamiltonian that achieves such a t^D -shaped light cone is non-generic and time-dependent in a rather artificial way. The Hamiltonians achieve the light cone because certain hopping terms are discontinuously turned off. Consequently, it is unclear to what extent the conclusions are reproducible for the actual Bose-Hubbard Hamiltonian (more below).

Two further comments on this:

- There is a possible interpretation of “optimal” as meaning “optimal within the wider class of time-dependent Bose-Hubbard Hamiltonians”. However, with the current abstract and first paragraphs, anyone who does not read the paper closely will be left with the impression that the bound is optimal for the standard, time-independent Bose-Hubbard Hamiltonian and that’s not what’s shown here. Moreover, the example Hamiltonian is discontinuous in time, which likely does not fall into the range of validity of the bounds (which is unspecified in the current manuscript).

Reply: Thank you for your comment! The Lieb-Robinson light cone is defined by the possible fastest protocol including the fine-tuning of the time-dependent Hamiltonian. This definition was originally adopted in the context of the Lieb-Robinson bound in long-range interacting (aka., power-law decaying interaction) systems, where the shape of the effective light cone is highly counterintuitive [M. Tran, et al., *Phys. Rev. X* 11, 031016 (2021)]. As the background for such a definition, the Lieb-Robinson bound characterizes the fastest case of information propagation; this is why the Lieb-Robinson bound characterizes the most universal constraints in many-body physics.

On the other hand, as you pointed out, the term “optimal” often depends on the situation that we are interested in. For example, the optimal shape of the light cone is known to be linear in the special cases where we consider the information propagation in an initially steady state [Kuwahara and Saito, *Phys. Rev. Lett.* 127, 070403 (2021)]. In this sense, we changed the title as follows without mentioning the optimality:

“Effective light cone and digital quantum simulation of interacting bosons ”

We rephrased the term “optimal” throughout the manuscript (i.e., title, abstract, introduction, etc.). We only utilize the term by making it clear that it is optimal under the condition that the arbitrary time-dependent Hamiltonians are considered. For example, we rephrased a sentence in the introduction as follows:

“Until now, we are far from the long-sought goal of characterizing the optimal forms of the effective light cones for the speed of i) and ii) under the condition that arbitrary time-dependent tunings of the Hamiltonian are allowed. ”

- The authors expect that the mechanism they identify for the special choice of time-dependent Hamiltonian will also lead to the same information propagation in the standard and physically relevant Bose-Hubbard Hamiltonian. However, this is not at all clear at this stage. First, one needs to make sure that the state with high-boson density \bar{n} on the “1D information path” indeed diffuses the bosons to an $O(1)$ density in time $\sim \bar{n}^{\frac{1}{D-1}}$. If this is true, then one can create initial states that form these 1D information paths. Then, one needs to check that before these bosons diffuse again, they can create sustained rapid information transport for reasonably large times. I believe that the best one can hope to prove from the construction given here is that there will be special initial states and intermediate state-dependent time scales on which information propagation can fill the t^D -shaped light cone. (The paper itself does not discuss any specific route to proving optimality for the usual Bose- Hubbard Hamiltonian.)

Reply: Thank you for your comment! As you pointed out, if we consider the usual Bose-Hubbard Hamiltonian with a simple initial state (e.g., the Mott state), it is quite natural to expect that the linear light cone should be proved and the acceleration no longer occurs. Also, for the time-independent Hamiltonian with repulsive interactions, the 1D information path should not survive for a long time. In the study of the bosonic Lieb-Robinson bound, however, such simple intuitions are extremely challenging to rigorously prove^{*6}.

For example, by using the reverse time evolution, we would be able to consider the situation that bosons automatically concentrate on a one-dimensional region (Fig. 2):

1. We first consider a quantum state $|\psi_0\rangle$ which has an information path on a 1D region.
2. We then evolve the state by a translation invariant Bose-Hubbard Hamiltonian, which yields a steady-state $e^{-iHt}|\psi_0\rangle$ after a long time.
3. We choose the state $e^{-iHt}|\psi_0\rangle$ as our initial state of interest $|\psi_{\text{ini}}\rangle$, which is expected to hold the low-boson density condition.
4. Considering the Hamiltonian $-H$, the time evolution $e^{iHt}|\psi_{\text{ini}}\rangle$ yields the boson concentration onto the one-dimensional path.

^{*6} Indeed, even for the homogeneous Bose-Hubbard model with the Mott state as the initial state, there have been no mathematical tools to treat the information propagation in general.

FIG. 2. In the process (a), the bosons diffuse and the state goes to the steady state with a low-boson density. By reversing the time evolution, we are able to induce the boson concentration (b). During this process, the acceleration of the information propagation can occur.

Importantly, the initial state $|\psi_{ini}\rangle$ is quite rare in the whole phase space, but we always have to exclude the possibility that the initial state of interest is not included in this class of states. In this situation, it might be possible to observe the acceleration of the information propagation in the above setup. The computational cost of the numerical simulation is quite huge and beyond the scope of this study, but we added this point in the outlook section. Also, as a possible scenario, we added the above protocol to the supplementary information.

“ Finally, it is intriguing to experimentally or numerically observe the supersonic propagation of quantum signals using a mechanism similar to that illustrated in Figure 4. In our protocol, we employed highly artificial boson-boson interactions. Then, a significant open problem is whether acceleration can occur even under time-independent Hamiltonians. We anticipate that the boson transport in the initial step can be achieved by reversing the time evolution (see Section XII of Supplementary Information). As for the second step in the protocol, Ref. [41] has already noted that the group velocity of the propagation front of correlations is proportional to the boson number at each site. Hence, we believe that the proposed acceleration mechanism can be realized within the current experimental setups. ”

The limitations of the optimality result described above are hard to find in the current presentation, especially towards the beginning of the paper. Readers will easily get the impression that the question of optimality is settled for the standard Bose-Hubbard Hamiltonian when in fact it is not. The authors present a reasonable mechanism for fast information propagation in an artificial toy model, but they do not show that it has similar implications for the standard Bose-Hubbard Hamiltonian. This paper has plenty of other interesting insights also without discussing optimality the way it is currently done.

Therefore, I suggest to reword the title, the abstract, and the introduction to make it clear that the information propagation bound is only shown to be achieved for a special choice of time-dependent Hamiltonian.

Recommendation. If the authors revise the presentation (including title, abstract, and outlook section) in a way that makes the constraints of the derived optimality result significantly clearer, then I strongly recommend the paper for publication in Nature Communications.

Reply: Thank you for the positive assessment of the publication. With all the revisions mentioned above, we believe that the modified version of the manuscript can be suitable for publication in Nature Communications.

Additional comments.

Below is a list of minor corrections and comments for the consideration of the authors.

1. When the authors revise the presentation of the optimality result, they should also compare the mechanism they identify for the time-dependent toy Hamiltonian with the mechanism found by Eisert-Gross for supersonic information propagation even in 1D.

Reply: Thank you for your comment! The Hamiltonian by Eisert and Gross effectively enhances boson hopping, where the hopping amplitudes are proportional to the position. Our mechanism relies on the tuning of the local boson numbers, where the speed of information propagation is enhanced by the local boson numbers under the existence of the boson-boson interactions. We added the following paragraph to the main manuscript.

We finally discuss a comparison with the mechanism proposed by Eisert and Gross [28]. In their model, the Hamiltonian effectively amplifies the hopping of bosons, with hopping amplitudes directly proportional to their positions. In contrast, our mechanism relies on dynamically adjusting the local boson numbers. The crucial aspect is that, in the presence of boson-boson interactions,

the speed of information propagation is enhanced by the local boson numbers.

Additional comments.

2. On p. 2 the sentence “The function $f(\{\hat{n}_i\}_{i \in \Lambda})$ includes arbitrary long-range boson-boson couplings.” can be easily misunderstood and should be changed to “The function $f(\{\hat{n}_i\}_{i \in \Lambda})$ includes **arbitrarily long-ranged** boson-boson **density** couplings.” The same applies to a similar sentence shortly after Result 1.

Reply: Thank you for your comment! Regarding this point, we have received a relevant comment from another reviewer. The constraints of the form of $f(\{\hat{n}_i\}_{i \in \Lambda})$ depends on the problems and summarized as follows:

1. For Result 1, no constraints have been assumed. So, even the non- k -local form of $\prod_{i \in \Lambda} \hat{n}_i$ is allowed. By taking the commutator with the boson number operator and the Hamiltonian, only the free-hopping term H_0 survives. So, for an infinitesimally small time, the boson-boson interactions cannot influence the time evolution $\hat{n}_i(t) \rightarrow \hat{n}_i(t + dt)$.
2. For Result 2, we assume that the interaction length in $f(\{\hat{n}_i\}_{i \in \Lambda})$ is finite range. But, the type of the functions is still arbitrary; for example, the interactions like $e^{\hat{n}_i \hat{n}_j}$ are allowed.
3. For Result 3, in addition to the finite interaction length, we assume that the $f(\{\hat{n}_i\}_{i \in \Lambda})$ is given by a polynomial with finite degrees and coefficients. This is necessary to implement the Hamiltonian dynamics by the interaction picture [Eq. 7 in the main manuscript]. In more detail, the finite degree polynomial with finite coefficients ensures Eq. 34 for the time derivative of the effective Hamiltonian.

These points have been summarized in Table II in Supplementary information. In the revision, we made these conditions more explicit in the main manuscript. We add the sentences as

“ $f(\{\hat{n}_i\}_{i \in \Lambda})$ is an appropriate function of the boson number operators $\{\hat{n}_i\}_{i \in \Lambda}$ with $\hat{n}_i = b_i^\dagger b_i$. The constraints on the function $f(\{\hat{n}_i\}_{i \in \Lambda})$ depend on the specific problems under consideration. These constraints are explicitly detailed in the statements of our main Results 1–3. In Result 1, there are no restrictions on $f(\{\hat{n}_i\}_{i \in \Lambda})$; in other words, arbitrary long-range boson-boson couplings are allowed. Result 2 requires a finite interaction length, but no additional constraints. In Result 3, alongside a finite interaction length, we assume that the form of the function is polynomial.”

Also, in each of the statements for the main Results, we clarified the condition for $f(\{\hat{n}_i\}_{i \in \Lambda})$.

3. On p. 2 when writing “Moreover, all results are applied to the time-dependent Hamiltonians.” the authors should briefly specify what kind of time dependence is allowed. (E.g., the coefficients are taken to be smooth functions in t .)

4. On p. 3, the authors write “As discussed in Result 2, the speed of information propagation is proportional to t^{D-1} .” This implicitly assumes optimality which is not proved in Result 2. The sentence should be changed to “As discussed in Result 2, the bound on the speed of information propagation is proportional to t^{D-1} .”

Reply: Thank you for your comments! On the time dependence, as in the case of the standard Lieb-Robinson bound, arbitrary time dependences can be considered. We added one sentence in the setup as

“Furthermore, similar to the Lieb-Robinson bound in spin/fermion systems, all our results are applicable to Hamiltonians with arbitrary time dependences.”

On the point 4., we rephrased the sentence as you proposed:

“As discussed in Result 2, the bound on the speed of information propagation is proportional to t^{D-1} .”

5. After eq. (5) a space is missing after “circuit”.

6. In the paragraph after (11), τ and t get mixed up a few times. I believe t should actually be replaced by τ in a few places.

7. On p. 6, the estimate $\text{tr}(\rho \hat{D}_{X[\ell]}) \approx e^{-O(\ell)} \text{tr}(\rho \hat{n}_X)$ does not seem quite correct. First, $e^{O(\ell)}$ is not good notation because $O(\ell)$ just means growing at most like ℓ , better would be $O(e^{-C\ell})$. (This confusion also appears in several places in the SM. This is normally resolved by writing Ω for lower bounds and O for upper bounds.) Second, it is not clear why this estimate should hold. This would be the case if the j -sum in the definition of \hat{D}_X was restricted to X but it is not, it goes over Λ . So what happened to the particles near the surface of $X[\ell]$? In the Appendix, it seems like one only gets decay after solving the optimization problem over $0 \leq \ell \leq r$. Finally, there’s a hat over D missing in the next line.

Reply: Thank you for your comments! On the points 5. and 6., we corrected the typos.

On the point 7., we corrected $\mathcal{O}(\dots)$ notations by $\Omega(\dots)$ notations appropriately. In the supplementary information, we revised them.

Also, on the estimation $\text{tr}(\rho \hat{D}_{X[\ell]}) \approx e^{-O(\ell)} \text{tr}(\rho \hat{n}_X)$, we adopted the quantum state ρ such that all bosons concentrate on ∂X . The main purpose of showing the example is to demonstrate that in the operator inequality

$$\hat{n}_X(\tau) \preceq \hat{n}_{X[\ell]}(\tau) \preceq \hat{n}_{X[\ell]} + c_{\tau,1} \hat{D}_{X[\ell]} + c_{\tau,2},$$

tuning of the parameter ℓ makes the RHS tight, which resolves the drawback from Eq. (11), i.e., $\text{tr}(\rho \hat{D}_X) \propto \text{tr}(\rho \hat{n}_X)$ that makes $\text{tr}[\rho(\hat{n}_X + c_{\tau,1} \hat{D}_X)] = [1 + \Omega(1)] \text{tr}(\rho \hat{n}_X)$.

In the revision of the manuscript, we make the point clearer as follows:

“Using the above operator inequality, we can resolve the drawback in Eq. (11) that originates from the concentration around the boundary ∂X . Here, instead of Eq. (11), we have

$$\text{tr}(\rho \hat{D}_{X[\ell]}) \approx e^{-\Omega(\ell)} \text{tr}(\rho \hat{n}_X)$$

for the quantum state ρ which has the boson concentration on the region ∂X . ”

Reply to Reviewer #3

#Summary

In the manuscript NCOMMS-23-00767-T, the authors derive bounds on the spread of information in bosonic lattice systems. These bounds are notoriously hard to derive and are in some cases impossible. The authors get around the known limitations by assuming the well-behaved nature of the initial boson number distribution. Unlike other works, they do not need to make an assumption that the initial state is steady. The "well-behavedness" of the boson number distribution is bootstrapped by showing that if it is valid at time t , it is also valid at time $t + \Delta t$ (albeit to a lesser degree).

The authors show 3 main results. The first concerns the spread of bosons in a lattice system when there are arbitrary numberconserving interactions. The authors show that this speed of boson propagation grows at most logarithmically in time.

The second result concerns the spread of —information— rather than the spread of bosons. In this setting, the authors make an additional assumption on the initial boson number distribution (namely that the probability of observing a certain number of bosons at each site decays at least subexponentially with boson number). In this context, they observe that the speed of information propagation is now dimension-dependent. Specifically, they find that the speed is proportional to t^{D-1} in D dimensions (up to logarithmic corrections). This is a surprising result, since fermionic and spin systems have no such dimension-dependence of the speed (at least for short-range interacting systems). Nevertheless, the authors show that this behaviour is actually tight, instead of merely being an artifact of a possibly correct but loose bound. Specifically, they show that there is a state transfer protocol that achieves this speed of information propagation (although, see below for my questions about the proof).

Lastly, the authors derive an analogue of the result by Haah-Hastings-Kothari-Low on the gate count of simulating the bosonic system in time. For a very general form of the bosonic interactions, they obtain an upper bound for simulating a system for time t . This upper bound scales as t^{D+1} , which the authors surmise is not optimal and can be improved to t^D .

Reply: We would like to thank you for the careful reading and the nice summary of our achievement. In the following, we show the point-by-point revisions to answer your concern, in particular,

the justification of our transfer protocol.

Evaluation

The paper is extremely well-written and gives a high-level overview of the techniques, the subtleties involved, and walks the reader through an easier version of the proof in the Methods section before relegating the more powerful version using the same techniques to the Supplementary Information. It also covers very well the ideas used by other papers preceding it and gives a very good feel for the techniques.

I do have one question about the tightness result of Result 2, which I hope the authors can explain/respond to my satisfaction in order to convince me of its correctness. Barring this potential issue, I believe the rest of the manuscript is correct to the extent that I have read it. The issue is the following.

In the section where they argue for the optimality of the light cone, the authors claim that they realise a CNOT operation in a time that scales as $1/(\bar{J}\bar{n}_t)$ or as $1/(\bar{J}^D t^{D-1})$. I found the justification for this in the Methods section incomplete. According to the authors' explanation, the operations in Eq. (27) are performed in two stages, one where they apply the free boson evolution to achieve $|N, 1\rangle \rightarrow |1, N\rangle$, and another where they transform the state $|1, N\rangle \rightarrow |0, N + 1\rangle$. In the first step, the time taken to achieve the transformation should be $\sim 1/(\bar{J})$ since one is transforming the entire creation operator at site 1 to a creation operator at site 2 (and is thus transferring all the bosons on site 1 to site 2). And because this takes time $O(1)$, even if there is a strong bosonic enhancement in the second step, the overall time taken is still $O(1)$. So I don't understand how the authors claim that the time taken for a CNOT scales inversely with the boson number as mentioned. I hope there is a reasonable explanation here that surmounts this difficulty.

Reply: Thank you for your positive evaluation of our work and the presentation of our manuscript. On the transfer protocol, we use three operations in total:

1. The operation to convert as $|N, 1\rangle \rightarrow |1, N\rangle$.
2. The operation to convert as $|1, N\rangle \rightarrow |0, N + 1\rangle$.
3. CNOT operation on the information path under the encoding of the quantum states on j th row as $|\bar{n}_t, \bar{n}_t\rangle_j \rightarrow |1\rangle_j$ and $|\bar{n}_t - 1, \bar{n}_t + 1\rangle_j \rightarrow |0\rangle_j$.

The first two operations are used for the collection of bosons onto the information path, while the third operation is used to send information on the path. As you pointed out, the necessary time to perform the first and the second operations, say Boson-Transfer-Operation, is at least $\mathcal{O}(1/\bar{J})$. Note that at this stage the protocol shows NO acceleration (i.e., boson transport has a FINITE speed). Hence, the overall time to implement the operations 1 and 2 is still $O(1)$. Here, we use half of the total time $t/2$ to perform this Boson-Transfer-Operation, which allows us to collect bosons within the region that are distant from the information path by the length of $\mathcal{O}(\bar{J}t)$. Therefore, the number of bosons at each of the sites on the information path is given by $\bar{n}_t \propto \mathcal{O}((\bar{J}t)^{D-1})$.

The acceleration occurs in the third operation, i.e., the CNOT operation (Eq. 28 in the main part of our manuscript). Due to the boson concentration on the information path, we can perform

the CNOT operation only by a time of $\bar{J}/\bar{n}_t \propto \mathcal{O}(\bar{J}^D t^{D-1})$, which is achieved by the Hamiltonian as in Eq. (29) of the main manuscript:

$$H = H_0 + h(\hat{n}_2 - \hat{n}_1)\hat{n}_3 + U(\hat{n}_3\hat{n}_4 + \hat{n}_4 - \bar{n}_t). \quad (\text{R.11})$$

By using the remaining half of the time of $t/2$, we can perform the $(t/2)/(\bar{J}/\bar{n}_t) \propto t^D$ CNOT operations. This allows us to achieve the effective light cone as $R \propto t^D$.

We are concerned about our understanding of the point you're trying to convey. If possible, we would appreciate it if you could clarify the matter more precisely.

Minor Comments

— While the figures are very illuminating, Fig. 1 is not referred to in the main text.

— On page 1, the authors refer to two conditions for getting Lieb-Robinson bounds as (i) and (ii). On page 2, they refer to two primary targets i) and ii) separately. In the last line of the first paragraph of page 2 beginning "Until now, we are far from the long-sought goal of characterizing the optimal forms of the effective light cones for the speed of i) and ii)", it is unclear that they are referring to the primary targets and not the two conditions introduced on page 1. Please use other letters such as (a), (b) to avoid confusion.

Reply: Thank you for your comment! On the Fig. 1, we made the following sentences:

Figure 1 summarizes the main results, providing qualitatively optimal effective light cones for both the transport of boson particles and information propagation.

Also, we have adopted the labeling (a) and (b) for the basic conditions for the Lieb-Robinson bound.

— Just after Eq. (1), the authors say "The function $f(\{\hat{n}_i\}_{i \in \Lambda})$ includes arbitrary long-range boson-boson couplings". Please say whether f is also arbitrarily long-range and whether there is any bound on the strength of f . Please also give some intuition why the arbitrary long-ranged nature of f does not matter. I suspect it is because it commutes with the boson number but a clarification would be nice. Is a non- k -local interaction, for example of the form $\prod_{i \in \Lambda} \hat{n}_i$ (the product of all boson numbers) also allowed?

— In Result 1, please again specify whether the constants depend on f or some notion of the strength of f in some way.

Reply: Thank you for your comment! Regarding the constraints on the function f , we can summarize as follows:

1. For Result 1, no constraints have been assumed. So, even the non- k -local form of $\prod_{i \in \Lambda} \hat{n}_i$ is allowed. By taking the commutator with the boson number operator and the Hamiltonian,

only the free-hopping term H_0 survives. So, for an infinitesimally small time, the boson-boson interactions cannot influence the time evolution $\hat{n}_i(t) \rightarrow \hat{n}_i(t + dt)$.

2. For Result 2, we assume that the interaction length in $f(\{\hat{n}_i\}_{i \in \Lambda})$ is finite range. But, the type of the functions is still arbitrary; for example, the interactions like $e^{e^{\hat{n}_i \hat{n}_j}}$ are allowed.
3. For Result 3, in addition to the finite interaction length, we assume that the $f(\{\hat{n}_i\}_{i \in \Lambda})$ is given by a polynomial with finite degrees and coefficients. This is necessary to implement the Hamiltonian dynamics by the interaction picture [Eq. 7 in the main manuscript]. In more detail, the finite degree polynomial with finite coefficients ensures Eq. 34 for the time derivative of the effective Hamiltonian.

These points have been summarized in Table II in Supplementary information. But, in the revision, we made these conditions more explicit in the main manuscript. We add the sentences as

“ $f(\{\hat{n}_i\}_{i \in \Lambda})$ is an appropriate function of the boson number operators $\{\hat{n}_i\}_{i \in \Lambda}$ with $\hat{n}_i = b_i^\dagger b_i$. The constraints on the function $f(\{\hat{n}_i\}_{i \in \Lambda})$ depend on the specific problems under consideration. These constraints are explicitly detailed in the statements of our main Results 1–3. In Result 1, there are no restrictions on $f(\{\hat{n}_i\}_{i \in \Lambda})$; in other words, arbitrary long-range boson-boson couplings are allowed. Result 2 requires a finite interaction length, but no additional constraints. In Result 3, alongside a finite interaction length, we assume that the form of the function is polynomial.”

Also, in each of the statements for the main Results, we clarified the condition for $f(\{\hat{n}_i\}_{i \in \Lambda})$.

- There are typos in Kothari's name (multiple instances).
- Page 5, "The gate complexity of the quantum simulation has room for qualitative improvement from $|\Lambda|t^{D+1}\text{polylog}(|\Lambda|t/\epsilon)$ to $|\Lambda|t^D\text{polylog}(|\Lambda|t/\epsilon)$: what is this expectation based on? I don't know if there are any lower bounds known, are there?"

Reply: Thank you for your comment! We have fixed the typos as Kohtari \rightarrow Kothari.

Also, on the possibility to improve $|\Lambda|t^{D+1}\text{polylog}(|\Lambda|t/\epsilon) \rightarrow |\Lambda|t^D\text{polylog}(|\Lambda|t/\epsilon)$, the primary reason is that we rely on a looser Lieb-Robinson bound with the light cone $R \propto t^{D+1}$ (as shown in Sec. S VII in the supplemental).

In the derivation, we construct the effective Hamiltonian by uniformly truncating the boson number by $\bar{q} = \tilde{O}(t^D)$. But, this simple construction cannot yield a looser Lieb-Robinson bound instead of the optimal one with the light cone shape of $R \propto t^D$. If we could combine the refined analyses for the optimal light cone with the HHKL analyses, we might be able to derive the gate complexity of $|\Lambda|t^D\text{polylog}(|\Lambda|t/\epsilon)$.

In the revised manuscript, we rewrite the sentence as follows:

“Third, it is an interesting open question to seek the possibility of improving the current gate complexity $|\Lambda|t^{D+1}\text{polylog}(|\Lambda|t/\epsilon)$ and clarify the optimal gate complexity to simulate the quantum

dynamics. ”

- Page 6, in the definition of $\hat{\mathcal{D}}_X$, is the sum over j really over the whole lattice Λ ?
- In the equation after Eq. (11) (please consider numbering all the equations, since otherwise it is hard to refer to them), I believe the first argument should be τ instead of t .
- After Eq. (12), when the authors say ”By solving the above optimization problem appropriately”, it would be nice to say what the optimal r is. Or at least a mention of how it scales with problem parameters.

Reply: Thank you for your comment! First, the definition of $\hat{\mathcal{D}}_X$ is correct, i.e., $\hat{\mathcal{D}}_X := \sum_{i \in \partial X} \sum_{j \in \Lambda} e^{-d_{i,j}} \hat{n}_j$. When the j is separated from the boundary ∂X , the contribution of \hat{n}_j decays exponentially.

We also fixed the typo as $\hat{n}_X(t) \rightarrow \hat{n}_X(\tau)$ in the equation after Eq. (11).

On the optimal choice of $r \in [0, \ell]$, we can only ensure the existence of r to achieve the inequality (13). Hence, we cannot obtain any information on what the optimal r is. It originates from Lemma 13 in the supplemental. The proof is a bit tricky and we iteratively derive upper bounds for the optimized quantity. We added the following sentences in the Method section:

“We cannot solve the optimization problem (12) in general but ensure the existence of $r \in [0, \ell]$ that satisfies the following inequality (Lemma 13 and Proposition 16 in Supplementary information): ”

- Page 7, ”...the time evolution of the boson number operator \hat{n}_i is roughly upper-bounded by the boson number on the ball region $i[\ell_t]$, that is, $\hat{n}_{i[\ell_t]}$, where $\ell_t = O(t \log t)$ ”: what about the third term in Result 1 that goes as $(\dots + ts)^s$? Is it ignored here? But that is presumably not small always.
- Before Eq.(23), should it be $\hat{n}_j |\psi_{i,q}\rangle = 0$ instead of j in the subscript?

Reply: Thank you for your comments! On the first point, as you point out, we have taken in the leading term and ignored the non-leading terms for simplicity. In the rigorous calculations in the Supplemental, we have considered all the terms. In the revised manuscript, we have added one phrase on this point:

“where $\ell_t = O(t \log t)$, and we ignored the non-leading terms .”

On the second point, we rewrite the paragraph without introducing the quantum state $|\psi_{i,q}\rangle$.

Could you please see the reply below?

- I did not understand the point made in this paragraph and how the sentence "Therefore, the superposition of different boson number configurations prohibits the site-dependent boson number truncation" followed from the paragraph. It makes sense that taking $q_i \geq q$ is in some sense necessary. How does that prohibit a site-dependent boson number truncation?
- Second column of page 8: I got the matrix elements $\langle 1, N|V|1, N \rangle = +UN(N+1)$ (instead of a minus sign in front) and the same for the other matrix element.

Reply: Thank you for your comments! To make the first point clearer, let us consider a quantum state $|\psi\rangle$ that is given by the superposition of two states as

$$|\psi\rangle = a_1|\psi_1\rangle + a_2|\psi_2\rangle, \quad |a_1|^2 + |a_2|^2 = 1.$$

We also assume that the two states $|\psi_1\rangle$ and $|\psi_2\rangle$ have different boson number configurations \vec{q}_1 and \vec{q}_2 in the region L , i.e.,

$$\bar{\Pi}_{L,\vec{q}_1}|\psi_1\rangle = |\psi_1\rangle, \quad \bar{\Pi}_{L,\vec{q}_2}|\psi_2\rangle = |\psi_2\rangle.$$

Then, we expect that for the quantum state $|\psi_1\rangle$, the effective Hamiltonian $\bar{\Pi}_{L,\vec{q}_1}H\bar{\Pi}_{L,\vec{q}_1}$ gives a good approximation for the dynamics of $|\psi_1(t)\rangle$ for small t :

$$|\psi_1(t)\rangle \approx e^{-i\bar{\Pi}_{L,\vec{q}_1}H\bar{\Pi}_{L,\vec{q}_1}t}|\psi_1\rangle,$$

but this effective Hamiltonian does not give a good approximation for $|\psi_2(t)\rangle$:

$$|\psi_2(t)\rangle \not\approx e^{-i\bar{\Pi}_{L,\vec{q}_1}H\bar{\Pi}_{L,\vec{q}_1}t}|\psi_2\rangle,$$

since the boson configurations of \vec{q}_2 can be largely different from \vec{q}_1 . Conversely, we cannot utilize $\bar{\Pi}_{L,\vec{q}_2}H\bar{\Pi}_{L,\vec{q}_2}$ to approximate $|\psi_1(t)\rangle$, i.e.,

$$|\psi_1(t)\rangle \not\approx e^{-i\bar{\Pi}_{L,\vec{q}_2}H\bar{\Pi}_{L,\vec{q}_2}t}|\psi_1\rangle, \quad |\psi_2(t)\rangle \approx e^{-i\bar{\Pi}_{L,\vec{q}_2}H\bar{\Pi}_{L,\vec{q}_2}t}|\psi_2\rangle.$$

In this way, due to the superposition of quantum states with different boson number configurations, we cannot utilize the effective Hamiltonian $\bar{\Pi}_{L,\vec{q}}H\bar{\Pi}_{L,\vec{q}}$ for a specific boson configuration. We felt that the previous explanation was a bit hard to understand, and thoroughly rewrote the paragraph as follows:

"The primary challenge arises from the inability to obtain an accurate approximation for dynamics using the effective Hamiltonian $\bar{\Pi}_{L,\vec{q}}H\bar{\Pi}_{L,\vec{q}}$ for a specific choice of \vec{q} . This challenge is rooted in the superposition of quantum states with diverse boson configurations. For instance, consider a quantum state $|\psi\rangle$ represented as the superposition of two states, $|\psi_1\rangle$ and $|\psi_2\rangle$, where $\bar{\Pi}_{L,\vec{q}_1}|\psi_1\rangle = |\psi_1\rangle$ and $\bar{\Pi}_{L,\vec{q}_2}|\psi_2\rangle = |\psi_2\rangle$. Then, time evolution with the effective Hamiltonian $\bar{\Pi}_{L,\vec{q}_1}H\bar{\Pi}_{L,\vec{q}_1}$ provides a reliable approximation for $e^{-iHt}|\psi_1\rangle$ but not for $e^{-iHt}|\psi_2\rangle$. Conversely, time evolution with $\bar{\Pi}_{L,\vec{q}_2}H\bar{\Pi}_{L,\vec{q}_2}$ gives a good approximation for $e^{-iHt}|\psi_2\rangle$ but not for $e^{-iHt}|\psi_1\rangle$.

Thus, a specific boson number truncation using $\bar{\Pi}_{L,\vec{q}}$ cannot be applied uniformly to all superposed states. It is necessary to consider different boson number truncations depending on the boson configuration of the superposed states, as discussed in Section S VIII A of the Supplementary Information. ”

On the second point of $\langle 1, N|V|1, N \rangle = +UN(N + 1)$, we have fixed the typo.

REVIEWERS' COMMENTS

Reviewer #1 (Remarks to the Author):

In short, the authors have sufficiently addressed my concerns that I'd be happy to support publication (not that I was ever really opposed in the first place, more just on the fence). I would like to thank them for taking my comments under consideration, as their response has been illuminating for me as well.

Some further comments (none of which require any further actions from the authors):

- Regarding the extent to which these results can be generalized further, I do appreciate that there cannot be any completely general speed limit for bosonic systems, and in that sense the authors' results are already extremely general. My concern is more just that someone might have a specific type of Hamiltonian in mind which doesn't fall into the Bose-Hubbard form — say they're interested in interactions that are local but change the internal states of the bosons — and in that case they'd still have work to do to derive speed limits (if any exist) for their system. But at this point, given everything else, I'd say this is a rather minor concern.

- Regarding the novelty of their technical analysis, the authors have convinced me that their methods indeed overcome some new challenges that I had not appreciated. So I happily withdraw this point. Furthermore, as I'm coming to better understand the distinction between particle and information transport, I do really think the separation that the authors find is very interesting and worth highlighting regardless of the technical analysis. I appreciate the extra helpful details that the authors have provided here.

- Regarding why the authors' information-propagation protocol can't be adapted to a particle-propagation protocol, thank you very much to the authors for clearing up this point! I hadn't appreciated that the indistinguishability of the bosons plays a key role. Highlighting this gives me a much better feel for why particle and information transport can behave so differently as well.

Again, thank you very much to the authors for all the work they put into addressing my various comments.

Reviewer #2 (Remarks to the Author):

In their thorough revision, the authors have addressed most of my concerns. In particular the subtlety of what "optimality/tightness" means is much more clearly discussed. The significant edits serve to further explain crucial points in the discussion and they make the text much more accessible to a wider readership.

The only (relatively minor) lingering confusion that I have concerns my question what kind of time-dependence in the Hamiltonian is allowed. Since the proof involves taking time derivatives, I think it is necessary to assume that the time-dependent coefficients have bounded derivatives in time, but perhaps I am missing something. This assumption would be relevant insofar as the information propagation protocol in its current form uses discontinuous coefficients. However, in the grand scheme of things, it is a minor technical question that perhaps should be reconsidered by the authors but does not affect my overall judgement of the paper.

In general, I reiterate the assessment from my original report: This is a landmark result which breaks new ground both physically and methodologically. Particularly the challenging argument to improve the Lieb-Robinson velocity from $t^{\{D\}}$ to $t^{\{D-1\}}$ requires handling the phenomenon of "superpositions of 'bad' states" which has never been done before in this context.

For these reasons, I strongly recommend this paper for publication Nature Communications.

Reviewer #3 (Remarks to the Author):

I thank the authors for their thorough responses. Some of my comments came from misunderstanding what the authors were saying, and the responses as well as the changes to the main text are satisfactory. I believe all the results are correct now.

I also concur with the very good suggestions offered by the other Referees and appreciate the response and edits offered by the authors to address them.

As mentioned in my previous evaluation, I believe that the manuscript is very well-written and makes important progress on the question of speed limits in the presence of interacting bosons. I have no hesitation recommending acceptance.

All the referees have recommended the acceptances, and most of the comments do not suggest specific revisions of the current manuscript. We take in the comment by the second referee of

“The only (relatively minor) lingering confusion that I have concerns my question what kind of time-dependence in the Hamiltonian is allowed. Since the proof involves taking time derivatives, I think it is necessary to assume that the time-dependent coefficients have bounded derivatives in time, but perhaps I am missing something. This assumption would be relevant insofar as the information propagation protocol in its current form uses discontinuous coefficients. However, in the grand scheme of things, it is a minor technical question that perhaps should be reconsidered by the authors but does not affect my overall judgement of the paper.”

To reflect the comment, we add the following sentence in the setup section:

“Although any time-dependences are allowed for Results 1 and 2, we need an additional condition on the norm of the derivative as in Ineq. (36) for the proof of Result 3.”